# Roles of Histone H2A Variants in Cancer Development, Prognosis, and Treatment

**DOI:** 10.3390/ijms25063144

**Published:** 2024-03-09

**Authors:** Po Man Lai, Kui Ming Chan

**Affiliations:** Department of Biomedical Sciences, City University of Hong Kong, Hong Kong SAR, China; pomanlai2-c@my.cityu.edu.hk

**Keywords:** epigenetics, chromatin, histone variants, variants, cancers, H2A, histone 2A variants

## Abstract

Histones are nuclear proteins essential for packaging genomic DNA and epigenetic gene regulation. Paralogs that can substitute core histones (H2A, H2B, H3, and H4), named histone variants, are constitutively expressed in a replication-independent manner throughout the cell cycle. With specific chaperones, they can be incorporated to chromatin to modify nucleosome stability by modulating interactions with nucleosomal DNA. This allows the regulation of essential fundamental cellular processes for instance, DNA damage repair, chromosomal segregation, and transcriptional regulation. Among all the histone families, histone H2A family has the largest number of histone variants reported to date. Each H2A variant has multiple functions apart from their primary role and some, even be further specialized to perform additional tasks in distinct lineages, such as testis specific shortH2A (sH2A). In the past decades, the discoveries of genetic alterations and mutations in genes encoding H2A variants in cancer had revealed variants’ potentiality in driving carcinogenesis. In addition, there is growing evidence that H2A variants may act as novel prognostic indicators or biomarkers for both early cancer detection and therapeutic treatments. Nevertheless, no studies have ever concluded all identified variants in a single report. Here, in this review, we summarize the respective functions for all the 19 mammalian H2A variants and their roles in cancer biology whilst potentiality being used in clinical setting.

## 1. Introduction

Genetic material in eukaryotic cells are organized into chromatin. The chromatin is often described as a polymer with nucleosome as the fundamental repeating unit, as described by the classic “beads on a string” model [1]. Each nucleosome consists of 147 base pair (bp) of DNA wrapping around a protein octamer, which is assembled from equal parts of the core histones H2A, H2B, H3, and H4 [2,3]. On both ends of the nucleosome at the DNA entry/exit site, linker histone H1 binds to the nucleosome and play a crucial role in the maintenance of higher-order chromatin structure. Alternations in chromatin structure are associated with change in gene expression modulated by mechanisms such as histone post-translational modifications [4], ATP-dependent nucleosome remodeling and replacement of core histones by variant subspecies by specific chaperones [5]. 

The expression and deposition of core histones are strictly coupled with DNA replication to fill the gaps behind the progressing replication fork. In the human genome, genes encoding for replication-dependent histones are organized into three gene clusters. The largest cluster, *HIST1*, consists of 55 genes that reside in chromosome 6 (6p21-6p22). *HIST2* and *HIST3* which contain six genes and three genes, located at chromosome 1q21 and 1q42 respectively [6]. 

Histone variants are paralogs of canonical histones. The difference between a variant histone and its canonical counterpart could range from several amino acids to large non-histone domains. Similar to canonical histones, histone variants are also subjected to post-translational modifications (PTMs) such as methylation and ubiquitination [7,8,9,10]. As a result, when canonical histones are replaced, the stability of nucleosomes where variants reside will be altered due to its different structural properties. Variants can further be grouped under two classes, homomorphous and heteromorphous, according to their extent of amino acid sequence changes [7,11]. A group of genes that have no homologs and are not shared among closely related species, known as *ORPHAN* genes, are quickly evolved and only known to be related to species-specific traits, have characteristics to constitutively expressed throughout the cell cycle and transcripts deposited before and after the S phase, are responsible for the expression of histone variants [12]. 

Contrary to canonical histone-encoding genes, histone variant-encoding genes contain introns and their mRNAs are polyadenylated [13]. Histone variant genes are expressed in a replication-independent manner and are deposited onto the chromatin by specific histone chaperones and remodelers [14]. Owing to the distinct structural properties of histone variants compared with their canonical counterparts, incorporation of histone variants to the nucleosome will alter the DNA-octamer interaction and hence the overall nucleosome stability. Consequently, histone variants are important players in fundamental cellular processes like DNA damage repair, chromosome condensation, transcription initiation and termination [10,15]. Interestingly, histone variants are mainly expressed in somatic tissues, but some have evolved and exhibit a high degree of variation across species to perform lineage-specific tasks, these variants are predominantly expressed in male germline for sperm packaging [16]. 

Among all the four classes of core histones, the histone H2A family has the largest number of histone variants with 19 in mammalian species reported to date. In the past decades, reports have revealed the potentiality and capability of H2A variants being the driving cause of cancer. Recently, an increasing number of studies demonstrated the presence and some newly identified, whilst overexpression, down regulation, and mutation of H2A variants in a variety of cancers. Indeed, these variants participate in different stages of cancer progression including migration, invasion, and metastasis. Additionally, the disbalance of variants also induced changes in epigenetic plasticity and sustained oncogene expression programme [16,17]. More importantly, growing evidence suggested that H2A variants may act as novel prognostic indicators and/or biomarkers for both early cancer detection and therapeutic treatments. However, future research is needed to fully explore the role or other unprecedented H2A variants that may play in cancer, as it is thought to be substantially more than what is now understood. Nevertheless, no studies have ever compiled all the found H2A variants in a single published report. Here, in this review, we take the initiative in providing an overview, summarizing the respective functions for all the 19 mammalian H2A variants, with focus on their role in cancers and their potentiality being used in the clinical setting as a therapeutic target. 

## 2. Classification of H2A Variants

H2A variants are categorized as homomorphous or heteromorphous based on the degree of dissimilarity from canonical H2A. Variants like H2A.1 and H2A.2, only differ from the canonical H2A by a few amino acids, are categorized as homomorphous. On the other end of the spectrum, heteromorphous variants carry extra domains on either the N- or C–terminal which result in distinct structure from the canonical histone [7]. Examples of heteromorphous variants are macroH2A (mH2A), H2A.Z and H2A.Bbd (Bar-body deficient) [18]. 

To visualize the subtle differences in amino acid sequences in homomorphous variants, the classic acetic acid/urea (AU) polyacrylamide gel electrophoresis (PAGE) methodology has been improved with the inclusion of nonionic detergents, Triton X-100, forming TAU PAGE. On the other hand, heteromorphous variants will either be separated by acetic acid-urea or sodium dodecyl-sulfate (NaDodSO_4_) polyacrylamide gel electrophoresis (SDS-PAGE) [11,18,19,20]. 

## 3. The 19 Histone H2A Variants

The H2A family, with 19 mammalian variants reported to date, is the most diverse histone family among the other core histones (Table 1). The amino acid sequences of the 19 histone H2A variants vary greatly from one another but remain highly conserved during evolution, indicating that they all carried out indispensable functions [17,21,22]. In contrast to the other three classes of canonical histones, H2A proteins have both N– and C–terminal tails. Typically, the C–terminal sequences of the H2A variants are the most diverse while the core area is conserved (Figure 1). In addition, the L1 loop on some of the histone H2A variants (H2A.Z.1 and H2A.Z.2) or acidic patch (H2A.Bbd) were found to be clearly different from, and some even vanished within their isoforms [23].

### 3.1. H2A.X—Good Helper in DNA Damage Repair 

H2A.X was first identified in the 1980s. It belongs to the heteromorphous class and is found in all eukaryotes [10,24]. It is highly conserved from yeast to human, with 143 amino acids in length and has a molecular weight of −15 kDa [24]. H2A.X is a well-studied variant that plays a critical role in preserving the genome integrity, through taking part in DNA double strand break (DSB) repairing processes to repair DNA damage induced by either chemical, ionizing radiation, or homologous recombination during meiosis [25]. One unique feature of H2A.X is the presence of the SQ motif in the C–terminal region. 

Mammalian H2A.X has two SQ motifs (human: TQ, SQ, mice: SQ, SQ) [26]. Upon DNA damage, Serine 139 (S139), which is within the SQE motif, is phosphorylated by DNA damage signaling protein kinases, for instance, Ataxia Telangiectasia Mutated (ATM), ATM-Rad3-related (ATR) and DNA-dependent Protein Kinase (DNA-PK) in the Phosphoinositide 3-kinase (PI3K) pathway, yielding Gamma-H2A.X (γ-H2A.X) [26,27]. Of note, S139 phosphorylation is not necessary for the deposition of H2A.X at DNA repair sites [28].

When DNA double strand break (DSB) is detected, H2A.X is recruited to the DNA damage site by the histone chaperone Facilitate Chromatin Transcription (FACT) [29]. Recent studies demonstrated that H2A.X associated factor contains FACT, DNA-PK, PARP1 and other proteins that are required for DNA damage repair. Phosphorylation of H2A.X by DNA-PK causes conformational changes in the nucleosome, FACT complex thereby primarily catalyzes the exchange activities for H2A and H2A.X. Alternatively, ADP-ribosylation of SPT16 (one of the subunits of FACT), which is driven by Poly (ADP-ribose) Polymerase 1 (PARP1), greatly hindered exchange activities [30]. Importantly, accumulation of H2A.X at repair sites will markedly reduce upon downregulation of SPT16. This illustrated that SPT16 is required to facilitate new H2A.X deposition at repair sites through binding towards Proliferating Cell Nuclear Antigen (PCNA) to the PCNA-interacting protein (PIP) box that it harbors [28,31]. 

With the deposition of new H2A.X to DSB sites, phosphorylation occurs and the formation of γ-H2A.X can be quickly distributed along the chromosome from DSB sites, spread either bidirectionally or asymmetrically depending on whether DSB is within the topologically associating domains (TADs) or at the TADs border respectively [32]. The spreading creates a DNA damage repair (DDR) platform by initiating a cascade and enlisting many DNA repair components, such as Mediator of DNA damage Checkpoint 1 protein (MDC1), Mre11-Rad50-Nbs1 (MRN) complex, p53 Binding Protein 1 (53BP1), and Breast Cancer Type 1 Protein (BRCA1) [33,34]. Upon completion of this platform, γ-H2A.X will be dephosphorylated by Wild type p53 Induced Phosphatase 1 (Wip1) or displaced by histone exchange [35].

Phosphorylation of H2A.X can also occur on tyrosine 142 (Y142) driven by William Beuren Syndrome Transcription Factor (WSTF) kinase in unstressed cells [36]. EYA1 and EYA3-mediated dephosphorylation of Y142 occurs at DSB to facilitate precise localization of DNA repair factors, which would otherwise result in the recruitment of apoptosis-promoting components [36,37,38]. 

Interestingly, H2A.X is not only recruited to the chromatin during DNA damage repair (DDR), it also replaces canonical histones in a DSB-independent manner. A recent chromatin immunoprecipitation sequencing (ChIP-seq) on mammalian cells revealed that H2A.X, prior to phosphorylation, already forms uniformly distributed clusters within the nuclear volume, and are especially enriched in sub-telomere regions and active transcription start sites (TSS) in replicating cells [39,40]. While in mouse pluripotent stem cells (PSCs), H2A.X aggregates around extraembryonic lineage genes [40]. Future research is needed to address this issue because the underlying mechanisms governing the distribution of clusters are yet elucidated. 

#### 3.1.1. Non-DDR Function of H2A.X

The functions of H2A.X are not confined to DDR. H2A.X is also involved in the regulation of mouse Embryonic Stem Cells (mESCs) proliferation independent of DDR (Table 2). Deposition of H2A.X to rDNA (ribosomal DNA) promoters of mESCs recruits NoRC (Nucleolar Remodeling Complex) to the chromatin to limit mESC proliferation by silencing rRNA transcription [41]. Furthermore, H2A.X knockout (KO) mice, albeit, shows genomic instability, impairment of DNA repair, growth retardation, immune deficiency and even infertility in male mice [42]. The incapacity of sex body formation by condensation of H2A.X-deficient spermatocytes from X and Y chromosome is proposed as the cause of infertility, leading to failure of initiating Meiotic Sex Chromosome Inactivation (MSCI) [43]. In addition, H2A.X can regulate the proper expression of Zygotic Genome Activation (*ZGA*) genes and Transposable Elements (TE) to allow the establishment of totipotency of mESCs [44]. In human, γ-H2A.X was determined to regulate the fate of human Pluripotent Stem Cells (hPSCs) and progenitor cells. If the level of γ-H2A.X is sustained, hPSCs will have an enhanced fate towards neural development while leukemic progenitors’ hematopoiesis will be suppressed and vice versa [45]. 

In non-mammalian eukaryotes, H2A.X also manifests its role in regulating zygotic cell division and development of early embryos. H2A.X-F, a H2A.X isoform found only in aquatic species such as *Danio* and *Xenopus*, is expressed exclusively in late-stage oocyte, eggs, and early embryos [46]. As our review only focuses on mammalian H2A variants, more detailed information on full eukaryotic histone variants should refer to other published research articles.

#### 3.1.2. H2A.X Role in Cancers–Tumor Suppressor 

The crucial role of H2A.X in DDR qualifies it as a tumor suppressor. Indeed, H2A.X has a postulated role in cancer progression and metastasis (Table 2, Table 3 and Table 4). In a p53-null background, heterozygous loss of H2A.X can ameliorate genome instability and incidence of cancer. Furthermore, homozygous loss of H2A.X (11q23) will cause cancer to arise earlier than its heterozygous counterpart. The suggested mechanism for this result is the failure of damage-induced foci formation and co-localize to pre-existing γ-H2A.X platform, hence lowering the DSB signaling efficiency. With that, H2A.X-deficient cells are more sensitive to DNA damaging agents and such H2A.X mutations are being observed in a number of cancers, such as sarcomas, brain tumor, neck squamous cell carcinoma and are predominantly prone to B and T cell lymphomas [47,48]. 

Overexpression of H2A.X is associated with several diseases, the most well-studied are ovarian cancer (OC) and breast cancer [49,50]. High levels of H2A.X are usually associated with upregulation of the PI3K/Akt (Protein kinase B)/Mammalian Target of Rapamycin (mTOR) pathway. This signaling pathway is used to maintain a balance for regular cellular behavior, while activated AKT is used to correlate with mTOR-phosphorylated positivity and is likely to contribute to tumorigenesis by promoting cell proliferation and protein synthesis [51]. With evidence, increased AKT pathways by overexpressing H2A.X assist cancer cells to escape from apoptosis and continuously obtain nutrients to sustain their proliferation [51]. Surprisingly, H2A.X is found selectively overexpressed in normal adjacent tissue (NAT) rather than its origin of malignancy in OC. Extracellular matrix remodeling and epithelial-to-mesenchymal transition (EMT) are observed in these NAT, suggesting that H2A.X may exert a role in helping NAT to obtain well-prepared via cross-talking with cancer microenvironment (Figure 2A) [52]. However, the underlying principles of how a high level of H2A.X in NAT can lead to EMT stays unclear. 

EMT in colon cancer has been associated with silencing or downregulation of H2A.X. This is consistent with the suggestion that H2A.X is a regulator of EMT. Low expression of H2A.X upregulated the expression of several key EMT-related Transcription Factors (TFs), Slug and ZEB1 significantly, these two TFs turn out to cooperate to induce EMT programme and invasion [53]. Moreover, hypoxic-dependent reduction in H2A.X abundance in hepatocellular carcinoma (HCC) will increase γ-H2A.X level, consequently promoting angiogenic activity and Epidermal Growth Factor Receptor/Hypoxia-Inducible Factor 1 alpha/Vascular Endothelial Growth Factor (EGFR/HIF-1α/VEGF) signaling pathway to gain nutrients (Figure 2B) [54]. 

Apart from the amount of H2A.X, the site of phosphorylation also influences cancer progression. Of note, S139 phosphorylation is vital for normal genome stability. However, newly identified phosphorylation in tyrosine 39 (Tyr39) is seen in a variety of cancers as well. Atypically elevated Tyr39 phosphorylation level positively correlates with the proliferation rate of cancer cells, poorer differentiation, higher histological grade, size of tumor, higher risk of metastasis, and lower survival rate [55]. As a result, these studies remarkedly illustrated that a proper expression and regulation of H2A.X securely safeguard our genome stability and function as an essential component in tumor suppressive mechanisms.

#### 3.1.3. H2A.X–A Potential Prognostic/Biomarker for Cancers 

The major role of H2A.X in DNA damage repair positions it as attractive candidate as a biomarker. In the past decades, after identifying γ-H2A.X as a universal marker for DSB, its mRNA and protein phosphorylation levels are highly valued and to be applied in the diagnosis of cancers and survival outcome prediction, which indicated a defect in DDR (Table 3). γ-H2A.X is continuously being identified in multiple cancers, including cervical cancer, colorectal cancer (CRC), melanoma and OC, hence its level is credible enough to assess the effectiveness of cancer therapies, like chemotherapy and radiotherapy nowadays [56]. γ-H2A.X therefore serves as an indicator to monitor the radiation and chemo-drug treatment-induced DSB in cancer patients, so as to evaluate patients’ radiosensitivity, together with how patients respond to the drug treatment and try to minimize the side effects caused by chemotherapy [57,58,59]. 

A widely used approach in vitro is to investigate the amount of γ-H2A.X induced in peripheral blood lymphocytes after ionizing radiation to predict the high risk or early detection of cancers, such as CRC and bladder cancer [60,61]. Usually, high or constitutive accelerated levels of γ-H2A.X point toward the severeness of cancer stage and tumor sizes. Not only limited to in vitro sensing, tissue-specific genotoxicity can also be measured upon the γ-H2A.X signals by immunofluorescence in vivo. Plappert-Helbig et al. revealed that staining of γ-H2A.X can surpass standard comet assay in identifying lesions caused by genotoxicants [62,63]. This may also provide a new platform for future DNA damage quantification in cancer types.

### 3.2. H2A.Z–Regulator of Gene Transcription

H2A.Z is another heteromorphous class of H2A variants that was identified within the same period as H2A.X. It is 128 amino acids in length and shares only 60% similarity with the canonical H2A. Heterozygous deletion of H2A.Z is embryonically lethal, underlining its importance in early embryo development [64]. H2A.Z plays a pivotal role in establishing an intricate pattern of gene expression necessary for proper early development in mammals [64,65]. H2A.Z is highly abundant and enriched in the pericentric heterochromatin of undifferentiated cells like mESCs, but not in differentiated mouse or human cells. H2A.Z has diverse functions ranging from transcriptional regulation to DNA replication, DNA repair, cell lineage differentiation, chromosome aggregation and neuronal development (Table 2) [66]. 

In mammals, H2A.Z localizes around transcription start site (TSS), centromere and enhancer region. Accumulating evidence suggested that distinct PTMs of H2A.Z give rise to a variety of functions. Acetylated H2A.Z enriched in promoter regions are known to participate in gene regulation [67,68,69,70]. TIP60-mediated acetylation of H2A.Z destabilizes the nucleosome to facilitate transcription and recruitment of regulatory factors [68,71]. H2A.Z found in the centromere and H3K27me3-marked facultative heterochromatin of the inactive X chromosome (Xi) are mono-ubiquitinated at the C–terminal (K119, K120, K121 and K125) and participate in transcriptional repression [72]. 

On the structural level, H2A.Z-containing nucleosome is less stable and has higher DNA accessibility compared with canonical nucleosome [73]. It is suggested to be the consequence of an extended acidic patch and L1 loop changes arose when it formed dimer with H2B and interacted with the whole nucleosome complex [74,75,76]. Recent reports revealed that the C–terminal tail of H2A.Z is required for mediating the DNA unwrapping up to 40 base pairs, while N–terminal tail modulates nucleosome gapping [77]. On the other hand, FRET analysis revealed a higher stability of H2A.Z-nucleosome, whilst formation of chromatin fiber [67,74,76,78]. The functional role of H2A.Z remains controversial with multiple contradictory results, as it can have both activate and suppressive functions in transcriptional regulation. A possible reason for such a dynamic functional role in H2A.Z mainly depends on first, its interaction with different variants partners within a nucleosome context and second, PTMs [79,80]. 

Two H2A.Z isoforms are being identified and are well-conserved in chordates, they are H2A.Z.1 (*H2AFZ*), and H2A.Z.2 (*H2AFV*), which differ from H2A.Z by 3 amino acids (14th amino acid: Threonine in H2A.Z.1, Alanine in H2A.Z.2; 38th amino acid: Serine in H2A.Z.1, Threonine in H2A.Z.2; 127th amino acid: Valine in H2A.Z.1, Alanine in H2A.Z.2.) (Figure 1). Although no significant structural differences have been reported between the two isoforms, their L1 loop exhibit structural variation. This is suggested with an evolutionary substitution in the 38th amino acid that specially located within the histone-fold domain (HFD) and the flexibility of the L1 loop in preventing to clash with canonical H2A in heterotypic nucleosomes [23,70]. Although the two isoforms share high similarity structurally, the functions in mammals are non-redundant. While H2A.Z.1 knockout is embryonically lethal; H2A.Z.2 knockout reduces cell proliferation promote apoptosis [67,70,80]. H2A.Z.1 and H2A.Z.2 have recently been reported they play independent, isoforms-specific and tissue-specific role in gene regulation and neuronal development. H2A.Z.1 even has a dual role in interacting with RNA polymerase 2 for transcription initiation and elongation [81].

H2A.Z.2.2 is a recently identified primate-specific variant of H2A.Z.2 [82]. H2A.Z.2.2 is expressed in all human tissues but is predominately found in human brain [82]. The features that set H2A.Z.2.2 apart from H2A.Z is a shortened C–terminal and 6 amino acids differences (Figure 1).

The spliced variants caused an up to date least stable nucleosome complex. A suggested reason for H2A.Z.2.2-nucleosome severely destabilize is due to a unique docking domain (lack part of H3/H4 binding domain) of this spliced variant due to its much shorter C–terminal. Of note, only when the 6 amino acids are present on the docking domain can weaken the chromatin association [83]. Majority of H2A.Z.2.2 is freely diffused in nucleus, only a limited amount is being incorporated into nucleosome via TIP60 and SRCAP chaperone in a regulatory manner. The already known function of H2A.Z.2.2 is limited to its ability in destabilizing nucleosomes, further details are yet to be explored.

#### 3.2.1. H2A.Z Functions Other Than Gene Regulation

Other than gene regulation, recent studies also pointed out that the detection of H2A.Z at centromere guided the localization of SUMOylated Heterochromatin Protein 1-Alpha (HP1α), which contributes to chromosome segregation and chromatic compaction [72,84,85]. H2A.Z also takes part in DNA repair process (Table 2). The deposition of H2A.Z is mediated by the Snf2 Related CREBBP Activator Protein (SRCAP) complex in mammals. Nucleosome Assembly Protein 1 (NAP1) and FACT also recognize H2A.Z-H2B dimer but do not show a high preference over canonical dimer. Recently, a human chaperone Acidic Nuclear Phosphoprotein 32 Family Member E (ANP32E) is reported and found preferentially bind to H2A.Z C–terminal docking domain to facilitate the deposition and removal of H2A.Z [86,87]. ANP32E and human Inositol auxotrophy 80 (INO80) are recruited to the site of DSB at early stages of DDR and evict H2A.Z upon completion of DNA repair [88,89,90,91]. In addition, H2A.Z has been shown in relation to cognitive function, memory processing and neuronal development as aforementioned by inhibiting its incorporation or depletion [92,93,94]. This coincides with the tissue-specific enrichment of H2A.Z in the brain tissues. Furthermore, H2A.Z is shown to be a key regulator in coordinating EMT by having a dual function in both activating and repressing epithelial or mesenchymal related genes in a depletion manner [95].

H2A.Z is involved in cell cycle regulation by indirect regulation of related genes [96]. ChIP-seq in mESCs revealed H2A.Z localization at replication origin in mammals [97]. H2A.Z also plays indispensable role in ESCs and cell lineage differentiation [98]. Depletion of H2A.Z in mESCs can lead to loss of pluripotency and premature differentiation. H2A.Z regulates gene expression in ESCs by facilitating the binding of pluripotency factors to lineage-specific genes promoters, and downstream recruitment of Polycomb Repressive Complex 2 (PRC2). For cell lineage differentiation, H2A.Z has been implicated in variety of lineages including, but not limited to, cardiomyocytes [99], melanocytes (by H2A.Z.2) [100], myoblast [101] and hematopoietic stem cells [102].

#### 3.2.2. H2A.Z Role in Cancers–Oncogenic Variants

H2A.Z is highly expressed in multiple cancers for instance, breast, colorectal, liver, lung, prostate, metastatic melanoma, pancreatic ductal adenocarcinoma (PDACs) and bladder cancer [103,104,105,106,107,108,109,110]. Overexpression of H2A.Z promotes cell proliferation in both estrogen receptor (ER) positive breast cancer and prostate cancer [111]. Other than that, microarray assays and immunostaining in breast tumor revealed high expression of the H2A.Z.1 isoforms, abundance of which correlated with lymph node metastasis and poorer survival prediction (Table 4).

The proposed action of H2A.Z in ER positive breast cancer involves upregulation of estrogen receptor alpha (ERα) and c-Myc to promote proliferation [112,113]. In short, the overall underlying molecular mechanism is that H2A.Z, ERα and c-Myc form a positive feedback loop in regulating estrogen receptor signaling. p400 complex assists H2A.Z incorporation into ERα target promoter in the presence of estrogen, followed by the recruitment of FoxA1 to target enhancers to facilitate target genes’ expression. Activated ERα will bind to H2A.Z gene promoter to increase H2A.Z variants expression. Interestingly, proto-oncogene *c-myc* will simultaneously be upregulated by ERα in response to estrogen treatment. In line with this, c-Myc is demonstrated to directly regulate and enhance H2A.Z expression level by binding to H2A.Z promoter, turns out subsequently upregulate the transcription of cell cycle genes such as *cyclin B2*, *cyclinA2* and *PCNA*, which correlates with increasing cell proliferation and division (Figure 3) [111,114].

H2A.Z.1 is implicated in liver tumorigenesis and metastasis. H2A.Z.1 modulates the expression of key genes in cell cycle and EMT in HCC and intrahepatic cholangiocarcinoma (ICC) [115,116]. H2A.Z.1 overexpression in liver cancer cells has been deciphered to suppress cell apoptosis and the expression of negative cell cycle regulators such as p21 and p27. On the other hand, genes involved in cell cycle progression, such as *CDK2*, *CDK4*, *CDK6* and Cyclin/CDK complex (*cyclin D1*) are upregulated by H2A.Z.1 overexpression, resulting in pRb hyperphosphorylation and the subsequent loss of its tumor suppressive activity. Furthermore, when H2A.Z.1 is depleted, its metastatic potential was observed with a decrease in chemoattractant-stimulated migratory response and fibronectin whilst an increase in E-cadherin, thereby promote EMT through Transforming Growth Factor–beta (TGF-β) signaling pathway [106,117].

H2A.Z.2 is implicated in the tumorigenesis of aggressive metastatic melanoma. It was postulated that H2A.Z.2 promote proliferation of melanoma cells by recruitment of Bromodomain-containing protein 2 (BRD2) and E2F1 to the promoter region of E2F target genes, where H2A.Z is known to occupy [118]. It works by controlling E2F target genes’ transcription activities in melanoma cells. Recent studies have revealed that H2A.Z.2 interacts with BRD2, hence H2A.Z.2 levels also raised concurrently [118]. BRD2 together with E2F1 bind to E2F target genes in a H2A.Z.2-deposition dependent manner, in turn promote proliferation of melanoma cells. Depletion of H2A.Z.2 resulted in cell cycle arrest in G1/S phase, accompanied by downregulation of cyclin E and cyclin A and Rb hypo-phosphorylation. In addition, PTMs of histone H4 and H2A.Z.2 are also partly involved in driving aggressive melanoma. Hyperacetylation of these two histones enhance the interaction with BRD2, leading to downstream activation of E2F target genes to promote cell proliferation [70,118].

It is important to note that all three isoforms of H2A.Z (H2A.Z.1, H2A.Z.2, H2A.Z.2.2) are highly expressed in PDAC cell lines. Depletion of H2A.Z isoforms resulted in similar cell cycle arrest in G2/M phase [108]. Although all three isoforms have been implicated in the tumorigenesis of PDAC, overexpression of H2A.Z.1 and H2A.Z.2 are more potent activators of oncogenic phenotype, which is consistent with the role of H2A.Z.1 and H2A.Z.2 in cell cycle progression.

Besides melanoma and PDAC, acetylation of H2A.Z at TSS of active genes can lead to oncogene activation in prostate cancer. On the contrary, deacetylation of H2A.Z at TSS will cause tumor suppressor genes silencing [119]. Furthermore, a recent report has revealed that acetylated H2A.Z can generate new ectopic enhancers, resulting in aberrant gene transcriptional activation in androgen receptor (AR) dependent prostate cancer [120].

Collectively, these findings strongly support the oncogenic properties bared by H2A.Z and its isoforms (Table 2). Given that it takes part in multiple essential biological pathways, not only in gene transcriptional regulation but also in orchestrating DNA repair, chromatin compression, memory processing and even cell lineage commitment, which if their regulations either being disturbed or altered, tumorigenesis can be progressed.

#### 3.2.3. H2A.Z–Potential Diagnostic Biomarker and Therapeutic Target for Cancers

H2A.Z overexpression has been reported in other cancers and are associated with poor prognosis (Table 3). The observation that H2A.Z depletion can induce cell cycle arrest and apoptosis positions it as a potential therapeutic target. Proof-of-concept experiments have been carried out in ICC and melanoma cell lines. In ICC where high H2A.Z expression predicts for poor survival, H2A.Z depletion sensitized ICC cell lines to cisplatin treatment. Cisplastin treatment followed by H2A.Z depletion resulted in apoptosis, inhibition of cell proliferation, and attenuated EMT [115]. Similar findings were reported in melanoma cells, where H2A.Z.2 depletion sensitized the cells to targeted therapies and improved drug efficacy [109]. Nevertheless, PDAC cell lines depleted of H2A.Z remain resistant to gemcitabine, this suggests that the action of drugs varies and highly depends on the type of cancers [108].

H2A.Z is also a potential prognostic indicator in cancer in addition to a novel therapeutic target. A fascinating viewpoint suggested that H2A.Z epigenetic signals together with other markers, such as 5MC, H3K9Ac and more were being detected in serum levels of circulating cell-fee nucleosomes (ccfn). By measuring levels of H2A.Z will indeed be crucial for early detection of cancer like CRC and PDAC [104]. Moreover, in HCC patients, H2A.Z.1 is further capable in regulating immune infiltration and T cell differentiation. Therefore, the measuring of both H2A.Z.1 expression and TP53 status may provide a more concrete prediction of HCC survival and a comparatively precise treatment to corresponding patients: high H2A.Z.1 are more likely to be sensitive towards immune checkpoint blockers (ICBs), while low H2A.Z.1 will have a better outcome with radiotherapy and chemotherapy [121]. Altogether, this fully manifests that H2A.Z is a promising diagnostic biomarker and a therapeutic target for cancers.

### 3.3. MacroH2A–Maintainer of Nuclear Organization and Heterochromatin Architecture

MacroH2A was first discovered in 1992 from rat liver nucleosomes [122]. It is an idiosyncratic histone variant with a tripartite structure—an N–terminus region sharing 64% amino acid sequence identity with H2A (1–122a.a.), an unstructured but lysine rich H1-like linker region (123–160a.a.), and lastly, a globular macro domain which size is two times that of the histone fold region (HFR) (161–371a.a.) [122,123,124,125] (Figure 1).

To date, macroH2A1 and macroH2A2, two mammalian macroH2A proteins are reported and encoded respectively by *H2AFY* and *H2AFY2* genes. Notably, two more alternative spliced variants of macroH2A1 exist, they are macroH2A1.1 and macroH2A1.2. MacroH2A1.2 solely different from macroH2A1.1 in the macrodomain binding pocket arises from alternative splicing of a mutually exclusive exon 5, allowing macroH2A1.1, but not macroH2A1.2 the capacity and implications for NAD+ metabolism and ADP-ribose signaling [126]. Despite a high homology between the two spliced variants, they have distinct function, which adds further complexity to the roles of macroH2A in different cellular processes.

In addition to playing a crucial role in maintaining nuclear organization and heterochromatin development, macroH2A also takes part in other molecular processes such as DNA damage repair and transcriptional repression (Table 2). It acts similarly to a plant-specific variant, H2A.W, in shielding approximately 10bp of extra-nucleosomal DNA from being digested by exonuclease 3. This protecting action is mediated by the interaction of its linker region with the DNA sitting close to the entry or exit site [75,127]. MacroH2A plays vital role in the structural maintenance of heterochromatin and nucleolar and is therefore concentrated at constitutive heterochromatin and repetitive DNA sequences. MacroH2A maintains heterochromatin architecture by promoting contacts between heterochromatic repeat and Lamin B1. Depletion of macroH2A resulted in heterochromatin de-condensation and irregularly shaped and expanded nucleolar. It is believed that the change in nucleolar morphology is a result of a surge in transcriptional activity of ribosomal DNA [128,129].

It is suggested that macroH2A contributes to genome-wide silencing of the X chromosome, as macroH2A (including isoforms) was commonly found to be concentrated on the inactive X chromosome (Xi) of female mammalian cells referred to as a macro chromatin body (MCB). Localization of macroH2A to Xi is dependent on X inactive specific transcript (*XIST*) lncRNA and two distinct macro-chromatin domains on macroH2A itself [130,131]. X chromosome inactivation (XCI) requires the maintenance of a repressive status, which was completed through several repressive mechanisms, including the deposition of macroH2A-H2B dimers into nucleosomes by Acidic Nuclear Phosphoprotein 32B (ANP32B) histone chaperone [132,133]. Several studies have revealed that heterotypic macroH2A-H2B dimer contribute to a more stable octamer formation, this tends to impede transcriptional activity by propagating into a higher-order chromatin structures, turn out assists in XCI [134,135]. However, macroH2A is non-essential for the initiation of XCI, as the process remains intact in macroH2A knockout mice [136,137].

All macroH2A isoforms are likely engaged in heterochromatin maintenance, each with its distinct role. MacroH2A1 is found predominantly at methylated CpG island within facultative heterochromatin, such as that on the Xi [138]. MacroH2A1 localization to the Xi is dependent on the ubiquitin E3 ligase Cullin3^SPOP^ (Speckle-type POZ (pox virus and zinc finger protein) protein) complex, demonstrated by the loss of Xi localization upon Cullin3 knockdown. Prominent X chromosome reactivation was observed in macroH2A1-depleted cells in the presence of DNA methylase and/or histone deacetylase inhibitor, suggesting that macroH2A1 is part of the epigenetic mechanism required for stable silencing of Xi [139,140,141]. For macroH2A2, evidence also revealed its presence on Xi and its potential role in shaping the chromatin organization in both compacted and accessible chromatin in regional contexts, through modulating as its cognate enhancer element [142,143]. On the other hand, the spliced variants macroH2A1.1 and macroH2A1.2, are present in a diffuse manner within the nucleus, and the formation of MCBs on the Xi is in a XIST-dependent manner [144]. Interestingly, macroH2A1.2 has been shown to inhibit Poly(ADP-ribose) Polymerase 1 (PARP-1 ) enzymatic activity, turn out perturbs the silencing effect of PARP-1 on chromatin [145]. Despite multiple reports on macroH2A1.1 enrichment on Xi, its contribution to the stabilization and maintenance of facultative heterochromatin structure remains obscure.

#### 3.3.1. Other Roles of MacroH2A and Its Isoform

Beside heterochromatin maintenance, macroH2A and its isoforms are engaged in DDR, transcriptional regulation, cell fate differentiation, mitochondrial function, and energy metabolism. MacroH2A1.1 is recruited to DSB sites by its interaction with the PARP1, which then facilitates the recruitment of 53BP1 together with Ku heterodimer: one of the DNA repair factors (KU70/80) for Non-Homologous End Joining (NHEJ) and assists in the activation of Checkpoint Kinase 2 (CHEK2) [146,147]. Unlike macroH2A1.1, macroH2A1.2 that lacks a macrodomain binding pocket, is known to mediate Homologous Repair (HR) through recruitment of BRCA1 [75,148].

MacroH2A not only has a well-known function in gene repression, it also has a dual function in signal-induced gene activation. One of the best examples is macroH2A1, but not macroH2A2, regulates memory processing and formation, which its occupancy in hippocampus is dynamically modified in order to promote learning-induced transcriptional activation, especially in upregulated genes following training [149].

MacroH2A1.1 can also sustain mitochondrial function and energy metabolism by optimizing NAD+ level [150]. Furthermore, macroH2A2 acts as a reprogramming barrier in fully differentiated cells, as it is enriched in early pluripotency genes marked by H3K27me3, which prevent reactivation of these genes [151].

Collectively, macroH2A and its isoforms displace a wide range of functions in a variety of cellular processes and give rise to isoforms-specific functions (Table 2).

#### 3.3.2. MacroH2A Roles in Cancer–Tumor Suppressor

Recently, there has been a surge in interest regarding the role of macroH2A and its isoforms in cancer tumorigenesis. There are strong evidence supporting the role of macroH2A and its isoforms as a tumor suppressor. Low expression of macroH2A has been reported in multiple cancers, such as melanoma, breast, colorectal, liver, lung, bladder, cervical and ovarian cancers [152,153,154]. When the level of macroH2A is restored, cancer cell proliferation, migration and metastasis are alleviated, reducing the malignant phenotypes induced by depletion of macroH2A (Table 4).

MacroH2A1.1 is implicated in EMT suppression as recent reports demonstrated dramatic decrease in macroH2A1.1 expression after the EMT induction in immortalized Human Mammary Epithelial Cells (HMLE). Ectopic expression of macroH2A1.1, but not macroH2A1.2, effectively inhibited EMT induction and HMLE exhibits less mesenchymal morphology. The underlying mechanism for the suppression of EMT is related to the presence of Poly (ADP-Ribose) (PAR) binding domain unique to macroH2A1.1. However, overexpressing macroH2A1 and macroH2A1.2 alone cannot reverse a fully mesenchymal cell to undergo Mesenchymal-Epithelial Transition (MET). Whether the reversible action of cells convert back to epithelial status is mediated through cooperating or interacting with other MET inducing signaling factors requires future investigation [155]. The depletion of macroH2A and its isoforms can also induce malignant melanoma progression through enhancing cell proliferation, migration and metastasis. Melanoma tumorigenesis can be suppressed through a direct transcriptional upregulation of a colorectal cancer oncogene, *CDK8*, with macroH2A and isoforms [154].

MacroH2A1.2 have been shown to inhibit osteoclastogenesis in bone metastases of breast cancer. This action can be inhibited by macroH2A1.2. It works by attenuating the expression and secretion of Lysyl Oxidase (LOX) by repressing the *LOX* gene through recruitment of EZH2, resulting in H3K27me3 deposition and ultimately gene silencing [156].

Furthermore, macroH2A1.2 can also inhibit prostate cancer induced osteoclastogenesis via direct interactions with HP1α and H1.2, turn out inactivating a major stimulator of osteoclastogenesis, Lymphotoxin Beta (*LTβ*) gene in prostate cancer cells. Altogether, this shows that even macroH2A1.2 does not play a role in suppressing EMT just as macroH2A1 and macroH2A1.1 do, it has a novel role in mediating bone metastases (Figure 4) [157].

In glioblastoma, the tumor-suppressive function of macroH2A2 is mediated through repression of genes associated with stemness and self-renewal. Depletion of macroH2A2 significantly promoted the self-renewal properties in cancer cells, while overexpression antagonized self-renewal [143,158]. In anal neoplasm, high macroH2A2 expression is associated with low-grade neoplasm and delayed recurrence. Interestingly, macroH2A2 loss in Human Papillomavirus (HPV) positive anal Squamous Cell Carcinoma (SCC) drives tumorigenesis. Of note, oncogene *E7* from HPV is vital for inactivating RB to activate E2F for cell proliferation. However, even in HPV-positive SCC samples, where macroH2A2 also functions as an inhibitor to reduce the expressions of E2F-regulate genes, E2F can be activated. Therefore, future work has to be performed to further elucidate the relationships between macroH2A2 and HPV status [159]. Last but not least, high expression of macroH2A2 can impede metastasis in Disseminated Cancer Cells (DCCs) by inhibiting cell cycle and oncogenic signaling programmes [160].

#### 3.3.3. MacroH2A and Isoforms–Novel Prognostic Factor and Diagnostic Markers

MacroH2A and isoforms expression levels highly correlate with patient survival prediction and are proposed to be a novel diagnostic marker in different types of cancers.

It is known that macroH2A1 and isoforms’ expression levels are usually high in different breast cancer types with worst prognosis (Table 3). In Human Epidermal Growth Factor Receptor 2 (HER2) positive breast cancer, macroH2A1.2 expression level is extremely high. MacroH2A1.2 has a trinucleotide insertion (-EIS-) sequences that is absent in macroH2A1.1, is responsible for the interaction and binding between HER2 receptors and its promoter [153]. MacroH2A1.1 contributes to the tumorigenesis of Triple Negative Breast Cancer (TNBC) primarily through modulating the EMT process. The high abundance of macroH2A1.1 has been linked to worse clinical outcome survival. Unlike other prognostic markers, macroH2A1.1 is suggested to be an independent factor that does not rely much on the proliferative status of cells [161,162].

Besides breast cancer, macroH2A1 also showed diagnostic and prognostic value in HCC and lung cancer recurrence. MacroH2A1 is upregulated in HCC and its expression has been associated with poor disease outcome. Gene Set Enrichment Analysis (GSEA) has revealed that high macroH2A1 expression is associated with upregulation of fatty acid metabolism and Mitogen-Activated Protein Kinase (MAPK) signaling pathways, which facilitate HCC tumor cells to better adapt to their microenvironment [163,164]. In lung cancer, macroH2A1.1 can serve as a biomarker for senescence in tumors, which is an anti-tumor mechanism. Patients with low macroH2A1.1 expression are more likely to experience recurrence, a correlation specific to macroH2A1.1 [152].

In anal neoplasm, low macroH2A2 expression is associated with high-grade anal cancer and early recurrence [159]. In contrast, patients with recurrence free survival used to express high level of macroH2A2. These findings suggest that macroH2A2 may serve as a biomarker for accessing the malignancy of anal cancer progression in order to have a more concrete outcome prediction and treatments.

To summarize, all isoforms of macroH2A generally act as a tumor suppressor, with evidence supporting their potential as both diagnostic and prognostic marker in multiple cancers.

### 3.4. Short H2A (sH2A)–H2A.R

Short H2A histone (sH2A) is a group of H2A variant which displayed a testis-restricted transcription pattern and are expressed exclusively during mammalian spermatogenesis [21]. sH2A is one of the most rapidly evolving group of histone variants and was originated from placental mammals. Features shared among sH2A variants include a truncated docking domain in the C–terminal, almost a complete loss of acidic patch and loss of lysine residues in the N–terminal. Due to the structure differences from canonical H2A, nucleosomes containing sH2A wrap less DNA, which results in loosely packed chromatin.

Recently, an undescribed variant, H2A.R, has been identified and suggested to be a common ancestor for all the known sH2A to date [165,166]. Compared with the previously known sH2A, H2A.R variants are more similar to canonical H2A due to its conserved docking domain in its long C–terminal [167]. To date, there is only one report verified the expression of H2A.R in opossum’s testis (one of the mammals), but not other tissues or organs [167]. Due to a lack of research on this ancestral H2A.R variant, little is known about their role in spermatogenesis and cancer progression. This may be a subject for future investigation.

### 3.5. Short H2A (sH2A)–H2A.Bbd

sH2A is further classified into 4 different types of variants. Among the four types of variants, Barr body deficient (H2A.Bbd), or H2A.B, is the most researched type of sH2A. H2A.Bbd only shares 48% similarity with canonical H2A. H2A.Bbd differs from H2A in the following ways: (1) The C–terminal tail and the last segment of the docking domain is missing in H2A.Bbd; (2) the N–terminal tail contains 6 consecutive arginine; (3) a more basic L2 loop (Figure 1). H2A.Bbd-nucleosomes contains only 118 bp of DNA, less than the 147 bp in a canonical nucleosome [168]. Studies have demonstrated that H2A.Bbd is found throughout the genome but, is usually associated with transcriptionally active genes on autosomes and the active X chromosome. Little to no H2A.Bbd is incorporated onto the Xi [131,169].

As a result of the missing docking domain in H2A.Bbd, the interaction between H2A.Bbd and H3 is weaker compared with that of H3 and canonical H2A. Therefore, H2A.Bbd incorporation into nucleosome often results in relaxed chromatin [170]. Moreover, H2A.Bbd alters the chromatin remodeling activity mediated by SWItch/Sucrose Non-Fermentable (SW1/SNF) complex [75,171]. It is also shown to enrich at the site of active genes involved in mRNA processing, DNA synthesis and DNA repair [172]. Deposition pattern of H2A.Bbd mostly overlaps with regions marked by H4 acetylation, indicating an active gene transcription region in the genome [173,174]. In addition to that, H2A.Bbd-H2B dimer is spontaneously exchanged within nucleosome with the assistance of a NAP1 histone chaperone [175,176]. Functionally, recent reports have shown that knockdown of H2A.Bbd resulted in an altered chromatin structure in male mouse’s post-meiotic germ cells, yet without affecting spermatogenesis [174,177,178]. It is believed that H2A.Bbd might assist in the transition from histone-based chromatin to protamine-based chromatin during the late stages of spermatogenesis.

#### H2A.Bbd Role in Cancers

Previous work showed that cells expressing H2A.B will have a shorter S phase and an accelerated sensitivity towards DNA damage, which these are associated with oncogenesis [165]. As sH2A comprised most of the commonly seen canonical H2A mutations in their wildtype sequences, including R29Q or R29F substitutions and a removal of E121 amino acid in the truncated C–terminal [179]. Therefore, this implicated that all sH2As have evolved an onco-histone features and are highly conserved even been through rapid evolution (Table 2).

H2A.B was found to be involved in a broad array of cancer types, where H2A.Bbd’s role in Hodgkin’s lymphoma (HL) was reported. Overexpression of H2A.Bbd positively correlates with cancer cell proliferation (Table 3). It is postulated that H2A.Bbd promote proliferation by enhancing ribosome production. H2A.Bbd containing nucleosomes are commonly found at the promoter of ribosomal genes, at the same time, in a close proximity with RNA polymerase 1, providing a loosen chromatin with highly accessible DNA to facilitate rDNA transcription. The role of H2A.Bbd in promoting ribosomal gene expression is evident from the downregulation of ribosomal gene upon H2A.Bbd depletion.

In addition to promoting rDNA transcription to enhance cell proliferation, H2A.Bbd also engages in the regulation of pre-mRNA splicing of ribosomal proteins, together with PTMs and HIF-1 pathways to establish HL phenotype [180]. Based on data set from Cancer Genome Atlas Program (TCGA), aberrant expression of H2A.Bbd has been reported in the following cancers: urothelial bladder carcinomas (BLCA), cervical squamous cell carcinomas, Uterine Corpus Endometrial Carcinomas (UCEC), endocervical carcinomas and Diffuse Large B-cell Lymphomas (DLBCLs). Among all cancer types, expression of the H2A.Bbd is the highest in DLBCLs (Table 4) [181].

The significance of H2A.Bbd overexpression in the aforementioned cancers is currently unknown. Given its nucleosome-destabilizing effect, it is hypothesized that overexpression of H2A.Bbd might open up the otherwise masked regulatory sequence for oncogenic transcription factors, thereby promoting tumorigenesis.

H2A.Bbd is proposed as a disease biomarker for HL and other cancers since it is commonly seen in a wide range of malignancies; however, its specificity and sensitivity have not been examined and require further research.

### 3.6. Short H2A (sH2A)–H2A.P, H2A.Q, H2A.L

H2A.P, H2A.Q, and H2A.L are the three remaining histone variants grouped under sH2A according to the most updated research [165,182]. H2A.Q is a newly discovered sH2A present in eutherian mammals. H2A.Q is encoded by a single pseudogene and has a deletion of 7 amino acids in the loop 1 and α-2 helix region. H2A.Q is the shortest eukaryotic histone variants described to date [167]. Interestingly, H2A.Q is only transcribed in the testis of dogs and pigs, but not in other tissues. However, neither human nor any primates’ tissues exhibit H2A.Q expression, suggesting that H2A.Q may not be playing any critical or has lost its role in human or primates.

H2A.P is encoded by a single gene reported so far. Not much detailed information on H2A.P is available until now, by only knowing all sH2As including H2A.P are originated on a portion of X chromosome since the common ancestor of eutherian mammals. Two essential conserved arginine residues that interact with the minor groove of DNA are deleted in H2A.P. Hence, H2A.P containing nucleosomes are expected to be less stable. Furthermore, the last 14 constrained amino acids are a distinctive property that sets H2A.P apart from other sH2A (Figure 1) [167]. Currently, little is known about the actual role that H2A.P plays in mammals, therefore further research is needed to fully uncover that.

H2A.L is the fifth sH2A identified to date. H2A.L is lost in human while conserved in mice. In mice, H2A.L is also known as H2AL1, and has two more isoforms, H2AL2 (H2A.L.2) and H2AL3 (H2A.L.3). The two isoforms differ from each other by no more than 12 amino acids [182,183]. The function of H2A.L.3 has not been documented in any studies so far, necessitating further research to uncover the mystery of this variant.

H2A.L.2-containing nucleosome interacts with only 130bp DNA as it shares the same features with other sH2A. Interestingly, H2A.L.2-nucleosome behaves similarly to H2A.Bbd nucleosome, inhibiting both Remodeling the Structure of Chromatin (RSC) and SWI/SNF nucleosome remodeling and mobilization [165,184]. H2A.L.2 favors histone retention on pericentric heterochromatin for spermatogenesis that occur in mice, likely due to the presence of its N–terminal arginine rich motif, that is capable for RNA binding and guide its localization to heterochromatin [185]. After that, the incorporation of H2A.L.2 recruits Transition Protein (TP) loading into nucleosome, and further drives TP-dependent protamine assembly for final sperm genome compaction (Figure 5) [186,187]. The importance of H2A.L.2 has been demonstrated through knock-out experiments, where the mice became sterile. Altogether, H2A.L.2 is important for male mouse fertility but is lost in human, suggesting that H2A.L.2 function may be replaced by other sH2A such as H2A.B or H2A.P, but more research is necessary to determine how credible this suggestion is.

### 3.7. H2A.22 (H2A.J)

H2A.J is a mammal-specific histone variant encoded by a *H2AFJ* gene located on human chromosome 12 [166,188]. This variant is specially concentrated in the chromatin of senescent cells, where it stimulates expression of genes in the inflammatory signaling cascade in reaction to DNA damage (Table 2). H2A.J differs from canonical H2A at amino acid 11 (A11V) and contains a potential SQ phosphorylation motif at its C–terminal at around the last 7 amino acids (Figure 1) [189]. With these alterations, H2A.J nucleosome is quite stable through enhancing DNA and histone interaction [190].

To be noted, the C–terminal tail of H2A.J is functionally critical for the expression of inflammatory cytokines/chemokines that belong to the Senescent-Associated Secretory Proteins (SASP) family, as inflammatory related genes are severely downregulated when H2A.J is mutated [191]. Genes that heavily depend on H2A.J for their expression typically have it deposited at the promoter and coding regions. Curiously, H2A.J was not commonly found at TSS regions. It is hypothesized that there may be an internal competition between H2A.J and H2A.Z.

As formerly described, H2A.J shows high abundance in senescent cells having proliferation arrest and associated with persistent DNA damage. It also appears to be accumulated in aging mice in a tissue-specific manner. Reports tested with Hair Follicle Stem Cell (HFSC) and Interfollicular Epidermal Cells (IEC) obtained from both young and old mice, after inducing DNA damage through irradiation, only old mice’s IEC and HFSC show significantly higher H2A.J accumulation [192]. Remarkably, H2A.J’s expression level is very low in proliferating cells, but in young mice, it is seen to be enriched, albeit to varying degrees, in specific organs such as kidney, brain and liver. This suggests that H2A.J may in some way displaying tissues or organs specific functions regardless of the state of cell senescence. To a certain point, H2A.J’s actual role in these organs at young age requires further investigation.

#### 3.7.1. H2A.22/H2A.J Role in Cancer

Inflammatory cytokines produced by senescent cells such as interleukin 1, 6, 8 (IL-1, IL-6, IL-8) are contributor of chronic inflammation. As chronic inflammation is known to promote pro-tumorigenic processes such as angiogenesis, proliferation, invasion, and metastasis, H2A.J might promote tumorigenesis via the upregulation of SASP [193].

H2A.J is highly expressed in a variety of carcinomas, especially in ER-positive breast and prostate cancers [189,194]. The role of H2A.J in breast cancer is by modulating the expression of estrogen and metastasis-associated genes, while *H2AFJ* tends to be hypo-methylated and overexpressed, suggesting that it might be an oncogene for the luminal B type breast cancer (Table 4). Yet, further functional studies are necessary to validate the role of this new candidate oncogenes in breast tumorigenesis. Other types of cancers that H2A.J are also involved in include Kidney Renal Cell Carcinoma (KIRC), aggressive melanoma, bladder and brain cancer. In contrast to luminal B breast cancer where H2A.J might act as an oncogene, high level of H2A.J is associated with better outcome in prostate cancer, bladder cancer and all other subtypes of breast cancer, yet withdrawing luminal A. On the other hand, high H2A.J levels in KIRC, brain cancer and aggressive melanoma is associated with poor survival [195]. These observations suggested that role of H2A.J in cancer is highly dependent on the cancer types.

It is interesting to uncover that H2A.J may also play a role in chemoradiotherapy (CRT) resistance in CRC. H2A.J expression is much higher in CRT-resistant CRC compared with CRT-sensitive CRC. It has been confirmed that H2A.J is involved in regulating two important pathways, Mitogen-activated Protein Kinase 7/Extracellular signal Regulated Kinase 5 (MAPK7/ERK5) and Human Immunodeficiency Virus (HIV), Negative Factor (Nef) pathway in CRC, together with the enrichment of several inflammatory pathways to increase CRT resistance in CRC patients (Figure 6) [196].

Furthermore, H2A.J drives radiotherapy and temozolomide (TMZ) resistance in glioblastoma as well. H2A.J is upregulated in mesenchymal type of glioblastoma, the most aggressive subtypes of glioblastoma. H2A.J expression in mesenchymal type of glioblastoma is associated with Proneural-Mesenchymal Transition (PMT) and activation of Tumor Necrosis Factor—Alpha (TNF-α)/Nuclear Factor Kappa-light-chain-enhancer of activated B cells (NF-κB) pathways. These interacting signal networks allow contact also between Signal Transducers and Activators of Transcription 3 (IL-6/STAT3) and Histone Deacetylase 3 (HDAC3) (Figure 6) [197]. The role of H2A.J in promoting TMZ resistance is affirmed by the downregulation of the aforementioned pathways upon H2A.J depletion. Moreover, upregulation of H2A.J in combination with HDAC3 have been associated to poor overall survival in glioblastoma [198]. A more comprehensive and thorough mechanisms by which H2A.J regulate these pathways to modify the PMT progression in glioblastoma should be further explored.

H2A.J has also been implicated in the development of resistance towards sorafenib in HCC [199]. *H2AFJ* and other hub genes are significantly upregulated in HCC. However, it is unclear how and if H2A.J expression correlate with disease outcome and overall survival prediction. In addition to knowing sorafenib resistance of HCC is due to gene upregulation, it also takes part in accelerating signaling pathways including PI3K/Akt and Janus Kinase (JAK) /STAT pathways. The other hub genes, such as Dynein Light Chain LC8-Type 2 (*DYNLL2*), *SH3* and multiple Ankyrin Repeat Domains Protein 2 (*SHANK2*), and Metastasis-Associated Protein 3 (*MTA3*) all contributed to the resistance by regulating metastasis and cell cycle checkpoint [200,201,202]. It is suggested that H2A.J may act similarly as in glioblastoma through upregulating IL-6/STAT signaling, EMT signaling and TNF-α/NF-κB pathways to control inflammatory response, cell proliferation, migration, anti-apoptosis, and survival response (Figure 6) [203].

Collectively, H2A.J is overexpressed in a variety of cancers, but its mode of action is highly context-dependent, which is evident from the contradicting correlation with overall survival/disease outcome in different types of cancer.

#### 3.7.2. H2A.22/H2A.J–Potential Biomarker and Indicator for Therapeutic Treatment

Although H2A.J alone has proven to be an ineffective prognostic marker in glioblastoma [198]. It is possible that the combination of H2A.J expression levels and other established markers such as O^6^-methylguanine-DNA-Methyltransferase (MGMT), could lead to a more accurate and sensitive prediction for therapeutic treatment outcomes, particularly for those patients who has developed resistance to chemotherapeutic agent.

H2A.J may also serve as a biomarker for senescent stem and aging skin cells. Antigen Kiel 67 (Ki-67) is a well-known marker for proliferating cells, this surface marker will decrease upon aging, while H2A.J expressing keratinocytes will be accelerated [204]. 53BP1 foci used to escalate in aging skin cell but it is difficult to identify discrete 53BP1 foci using immunostaining; in contrast, H2A.J accumulation is much easier to be detected [192]. As a result, H2A.J may become a potential biomarker for senescent cells with increased efficacy and sensitivity.

### 3.8. H2A.1 and H2A.2

H2A.1 and H2A.2 are the homomorphous group of histone variants with a slight difference in amino acid sequence compared with canonical H2A (Figure 1). H2A.1, also known as TH2A. Genes encoding for H2A.1 are located on chromosome 6p21-22 in human, and chromosome 17p11 in mouse. H2A.1 is a mammalian specific histone variant expressed in testis, oocytes and zygote. H2A.1 differs from the canonical H2A by 10 amino acids and has a preference for forming a dimer with H2B.1.

Co-expression of H2A.1 with the four Yamanaka factors (OKSM: OCT4, SOX2, KLF4 and MYC), induce an open chromatin that enhanced the generation of iPSC. Domain swapping experiment demonstrated that the L1 loop region of H2A.1 is essential for the reprogramming [205]. A similar phenomenon was seen in human as well, co-expression of H2A.1 histone chaperone, Nucleophosmin/Nucleoplasmin 2 (NPM2), with OSKM promoted the generation of iPSCs from naïve stem cell [206]. Remarkably, in zygote, depletion of maternal H2A.1 leads to a perturbation in paternal genome activation, indicating a significant role in genome reprogramming [207].

H2A.1 is uniformly distributed and commonly enriched on X chromosome and autosomes. It was found that H2A.1 can exert both transcriptional upregulation and suppression to genes on the X chromosome. In addition, H2A.1 is also found to be highly expressed in testis and is involved in spermiogenesis and early embryonic mitosis. The 127th amino acid of H2A.1 in mouse is found to be phosphorylated (pH2A.1) during early embryo mitosis and spermatogenesis [208,209]. This data suggested that pH2A.1 is involved in chromatin condensation, which occurs during the last stage of spermiogenesis. Interestingly, paternal pH2A.1 that is rapidly removed upon fertilization will reappear in the pericentromeric heterochromatin of the first mitotic zygote.

Compared with other H2A variants, reports on H2A.2 are limited. Despite the contrasting function of H2A.1 and H2A.2, they are often discussed together in literature. H2A.2 differs from H2A.1 by a few amino acids, two of them are being well-documented; 16th amino acid (H2A.1: Threonine, H2A.2: Serine) and the 51st amino acid (H2A.1: Leucine, H2A.2: Methionine) [210]. H2A.2 coding genes are resided on chromosome 1q21 in human and chromosome 2q34 in mouse [211].

The role of H2A.2 has been investigated in the neuron differentiation, embryogenesis, and aging [212]. Deposition of H2A.2 and H2A.1 is observed during postnatal development for the maturation of rat brain cortical neurons. These two histone variants undergo a dynamic change right after birth and during the postnatal period. At birth, H2A.1 shows a high abundance, whereas H2A.2 represents only 27% of the total H2A. However, H2A.2 expression surged during postnatal development and gradually replaced H2A.1 to become one of the major replication dependent variants (42% of total H2A) [213]. One study pointed out these 2 variants are ubiquitinated in the terminally differentiated cortex neurons [214]. Nevertheless, the precise function of the two histone variants in the differentiation of the neurons remain to be elucidated.

In addition to neuronal development, their expression levels are also being examined during liver cell differentiation, from the embryonic to the adult stage. Notably, H2A.2 will gradually increases, while H2A.1 level decreases in adult hepatocytes [215]. This suggested that H2A.1 functions in epigenetic reprogramming in embryonic stem cells and undifferentiated cells, but the exact role of H2A.2 is unclear.

#### H2A.1, H2A.2 Role in Cancer

H2A.1 and H2A.2 have been associated with hepatocarcinogenesis. Their expression levels vary at different stages of cancer progression. Interestingly, while H2A.2 is the predominant variant in normal and preneoplastic tissue, H2A.1 is overexpressed in HCC. The high abundance of H2A.1 coincides with the other studies that suggested H2A.1 opens up the chromatin structure to facilitate gene transcription, hence cell proliferation for hepatocarcinogenesis (Table 4).

These two histone variants share almost 98% identity. They are synthesized and deposited during the G1 and S phases when examining hepatoma cells [216]. However, a single amino acid variation between the two variants leads to subtle differences in histone-DNA interaction. This could potentially explain the functional heterogeneity between H2A.1 and H2A.2 in regulating malignancy-associated genes [211]. The H2A.1/H2A.2 ratio is important, for instance, if H2A.1 is overexpressed, reprogramming of hepatocytes could take place through hyperproliferating normal liver cells to return back to preneoplastic and neoplastic phases, which would further promote the development of hepatocarcinogenesis. These results are consistent with earlier studies showing the role of H2A.1 in epigenetic reprogramming.

Recent discovery of CpG island at the promoter region of H2A.1 and TSS-proximal region of H2A.2 suggests that the expression of the two variants might be regulated by different epigenetic factors during tumorigenesis [217]. H2A.1 and H2A.2 are also expressed in colon cancer, and they somehow appeared to have varying degrees of acetylation and methylation [218]. H2A.1 is hypo-methylated in embryonic cells while H2A.2 is hyper-methylated in HCC. This observation explained the phenomenon why H2A.2 is not overexpressed in HCC despite its high expression during preneoplastic stages. Moreover, these data also indicate hypo-methylation of H2A.1, which may facilitate the development and spread of tumors, as the cause of HCC [219,220].

Collectively, it should be highly appreciated that the dynamic expression patterns of these two histone variants contribute to distinct time points of development have different effects (Table 2). More studies are required to solve all the confusion including the unknown mechanisms and roles of both H2A.1 and H2A.2.

## 4. Conclusions and Perspectives

The histone H2A family is a group of nuclear proteins containing 19 variants. The role of these variants are mainly reported in mammalian species. Remarkably, all H2A variants are involved in safeguarding our genome integrity. Recently, new insights have additionally pointed out some unprecedented variants, including H2A.R and H2A.Q, which are further being grouped together. Furthermore, new alternative spliced forms of variants, such as H2A.Z.2.2 are just being identified and its function is only limited to destabilizing nucleosome complexes. Therefore, the detailed functional role of all the 19 mammalian H2A variants cannot be thoroughly discussed here due to a lack of supporting evidence, which serves a subject for later elucidation.

On the other side of the coin, many H2A variants have been implicated in cancer development. Majority of H2A variants contribute to tumorigenesis and cancer progression. Aberrant expression of histone variants leads to the upregulation of regulatory genes involved in cell proliferation, migration, EMT, invasiveness and angiogenesis; whilst downregulating tumor suppressor genes and inflammatory responses [221]. It is worth noting that the expression of some particular variants is associated with development of drug resistance in certain cancers. Whether other variants may also guide chemotherapy resistance in clinical settings requires further investigation. Last but not least, it is believed that many of the H2A variants have the potential to serve as prognostic indicators and/or effective biomarkers for cancers. However, the specificity and sensitivity of a variant to be used individually for cancer detection remain questionable. With that, we suggested in combination with other well-established markers, more credible diagnostics outcomes may be identified, hence more precise and improved predictions and therapeutics treatments can be proposed. The validity of this suggestion necessitates further research, pointing toward a new direction for future studies.

## Figures and Tables

**Figure 1 ijms-25-03144-f001:**
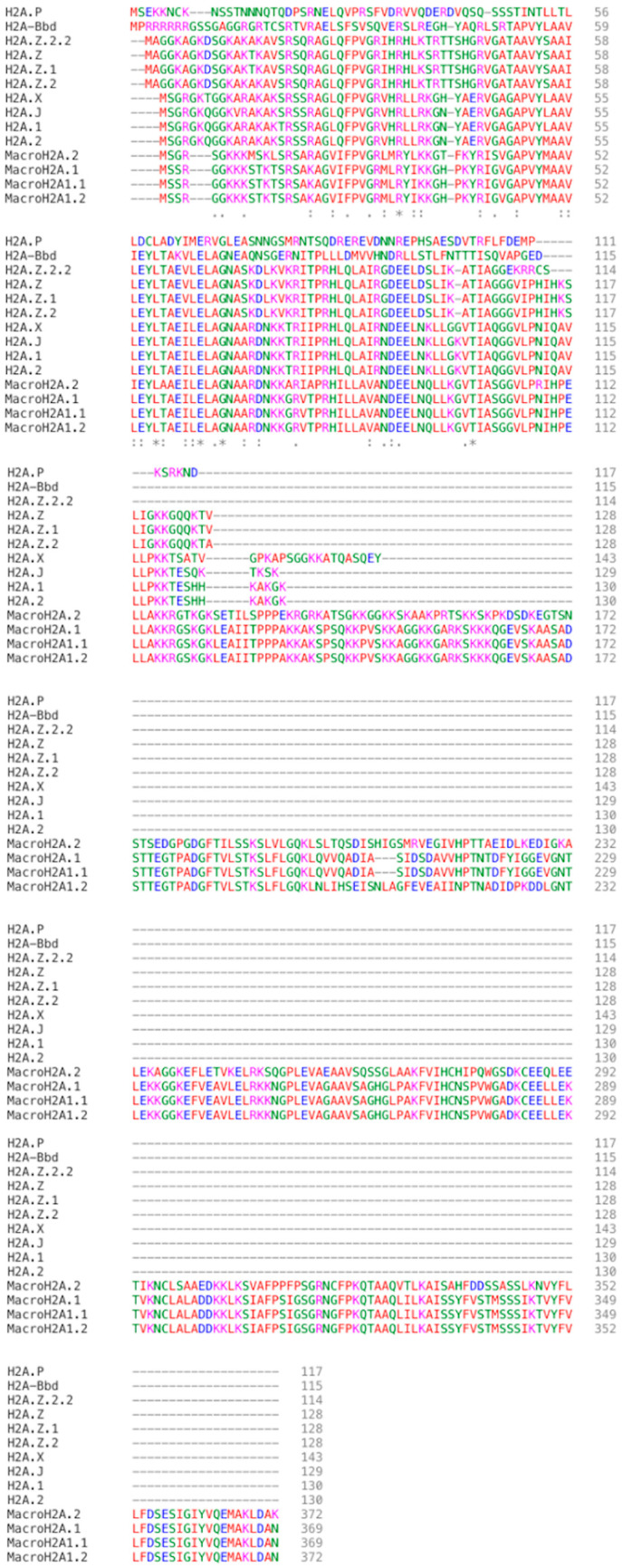
Proteins sequences alignment comparison of all the “*Homo sapiens*” found histone H2A variants. (GeneBank accession numbers: H2A.X “KAI4074518”; H2A.Z “NP_002097”; H2A.Z.1 “KAI2535323”; H2A.Z.2 “Q71UI9”; MacroH2A.2 “AF151534”; MacroH2A1.1 “AF044286”; MacroH2A1.2 “AF041483”; H2A.Bbd/H2A.B “AAL01652”; H2A.P “KAI3999235”; H2A.1 “P0C0S8”; H2A.2 “NP_003507”. Proteins sequences of H2A.Z.2.2 please refer to (Bönisch et al., 2012) published paper.

**Figure 2 ijms-25-03144-f002:**
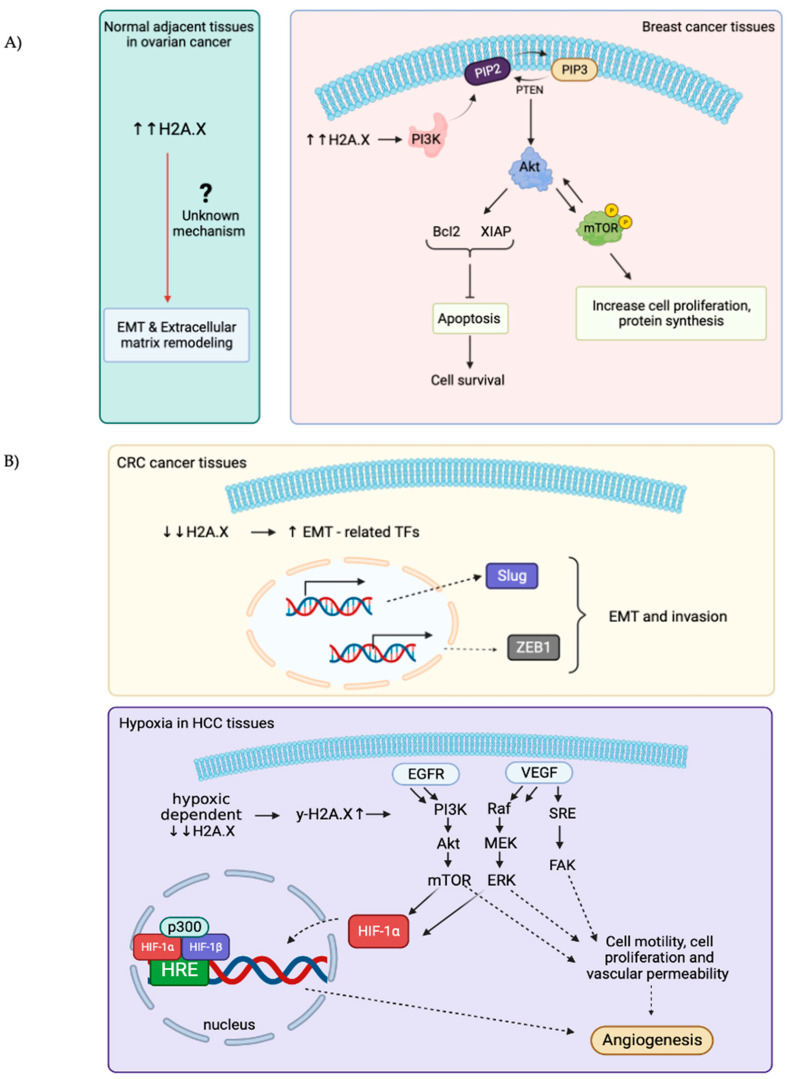
Schematic illustrations showing the role of overexpressed and down-regulated expressions of H2A.X in different types of cancers. (**A**): H2A.X is overexpressed in breast cancer and the normal adjacent tissues (NAT) in ovarian cancers. High levels of H2A.X used to activate downstream PI3K/Akt/mTOR signaling pathways to gain advantage for cell growth and cell proliferation in breast cancer. The underlying mechanisms for how the high H2A.X levels in NAT in ovarian cancer lead to EMT and extracellular matrix remodeling remains obscure. (**B**): H2A.X is downregulated in both CRC and HCC. Decreases in H2A.X in CRC can increase Slug and ZEB1 transcription factors expressions, which further enhance EMT and invasiveness of cancer cells. In HCC, H2A.X levels decrease under hypoxia condition, γ-H2A.X levels thereby increase and regulate multiples signaling pathways (EGFR/HIF-1α/VEGF) to sustain angiogenesis and nutrients uptake for cell growth and proliferation.

**Figure 3 ijms-25-03144-f003:**
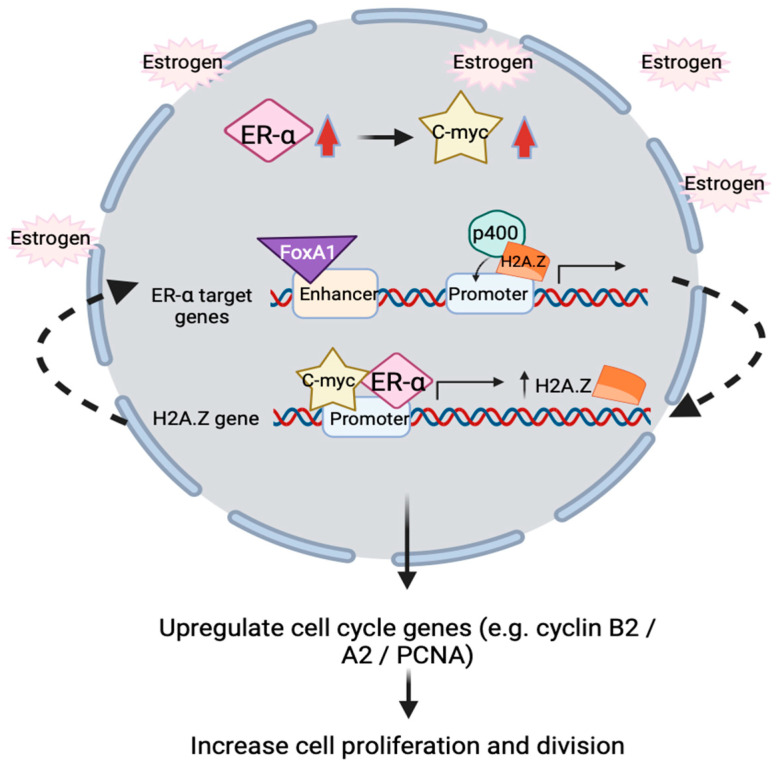
Schematic illustration showing positive feedback loop relationship between ERα, c-Myc and H2A.Z inside nucleus. In the presence of estrogen, ERα and c-Myc transcription factors (TFs) will simultaneously be upregulated. These two TFs will bind to H2A.Z gene’s promoter to enhance H2A.Z expressions. H2A.Z is incorporated into promoter of ERα target genes by the p400 complex, FoxA1 is then recruited to the enhancer of ERα target genes, resulting in upregulation of gene expression. This turns out upregulate the cell cycle genes and increase breast cancer cells proliferation and cell division.

**Figure 4 ijms-25-03144-f004:**
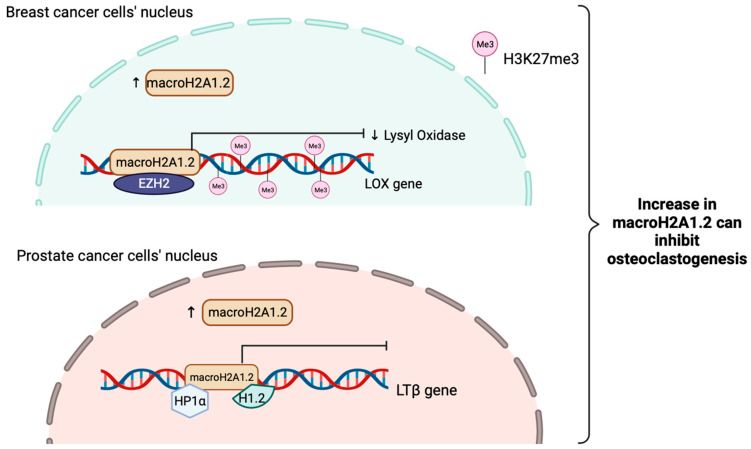
Schematic illustration showing the tumor suppressive role of macroH2A1.2 when is increased in breast cancer and prostate cancer in inhibiting osteoclastogenesis. MacroH2A1.2 will attenuate the expression of lysyl oxidase (LOX) and interact with EZH2 to raise the H3K27me3 levels to sustain gene silencing in order to inhibit tumor-induced osteoclastogenesis in breast cancer. In prostate cancer, macroH2A1.2 will directly interact with HP1α and H1.2 to suppress *LT**β* gene expressions, a key stimulator of osteoclastogenesis.

**Figure 5 ijms-25-03144-f005:**
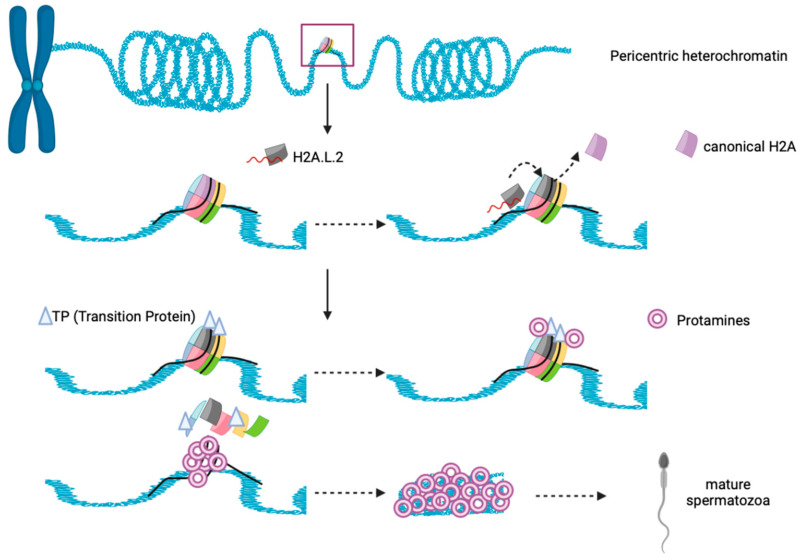
Schematic illustration showing functional interplay between histone, H2A.L.2, Transition proteins and Protamine during spermatogenesis in mice. N–terminal of H2A.L.2 allow the binding of RNA and it guides the histone variant to the heterochromatin’s nucleosome. Replacement of canonical H2A occurs and H2A.L.2 subsequently recruit transition proteins (TP) loading on nucleosome, and further drives TP-dependent protamine assembly and histone eviction to form mature spermatozoa.

**Figure 6 ijms-25-03144-f006:**
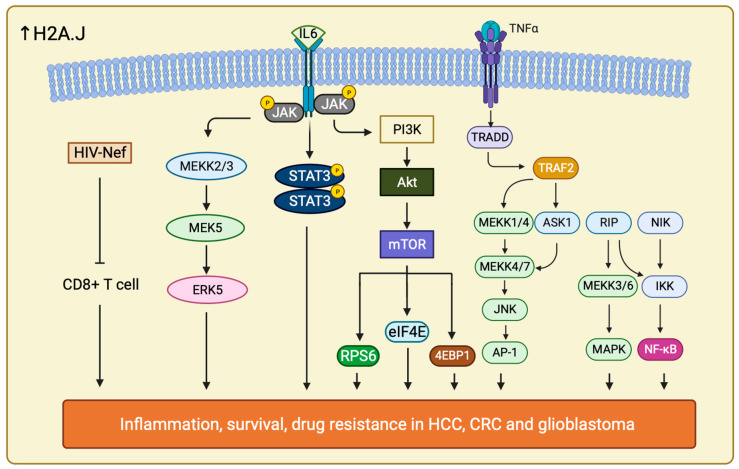
Schematic illustration showing all the signaling pathways that are involved in drug resistance in cancer therapy when H2A.J is overexpressed. Increase in H2A.J regulates multiple significant signaling pathways including ERK5, TNF-α/NF-κB, PI3K/Akt, JAK /STAT, HIV Nef, IL-6/STAT and more (not listed all here). These pathways play different role and allow cancer cells to survive against different drug treatments through accelerating large amount of inflammatory responses, escaping CD8+ T cell activity and enhance their cell proliferation rate and cell growth.

**Table 1 ijms-25-03144-t001:** Table showing H2A canonical histone and all the 19 mammalian H2A variants described to date in mammalian species.

Core Histone (Canonical Histone)	Histone Variants
H2A	H2A.X H2A.Z, H2A.Z.1, H2A.Z.2, H2A.Z.2.2 macroH2A: macroH2A1, macroH2A1.1, macroH2A.1.2, macroH2A2 shortH2A (sH2A): H2A.R, H2A.Bbd/H2A.B, H2A.L/H2A.L.1, H2A.P, H2A.Q H2A.L.2, H2A.L.3 H2A.22/H2A.J H2A.1, H2A.2

**Table 2 ijms-25-03144-t002:** Histone variants’ functions, alterations, and role in cancers. This table summarized all the 19 histone H2A variants’ primary and additional functions, alterations consequences and their role play in different cancer types.

Variants’ Name	Primary Functions	Other Functions	Role and Alterations in Cancers
H2A.X	DNA damage repair	1. Regulate and define mESCs proliferation 2. Immune response 3. Growth 4. Reproduction 5. Establishment of mESCs totipotency 6. Regulate fate of hPSCs and progenitor cells	Alterations’ consequences P53^−/−^ heterozygous loss: genome instability P53^−/−^ homozygous loss: early cancer formation, impairment of DNA repair, growth retardation, immune deficient and infertility Mutations observed in sarcomas, brain tumor, neck squamous cell carcinoma, B and T cell lymphomas Alterations in phosphorylation site of H2A.X (Tyr39) Role in cancer: Tumor suppressor Regulate cancer progression and metastasis (breast, ovarian cancers, HCC, and CRC)
H2A.Z	Regulator of gene transcription	1. DNA replication 2. DNA damage repair 3. Cell lineage differentiation 4. Chromosome aggregation 5. Neuronal development, cognitive function, memory processing	Alterations’ consequences H2A.Z^+/−^ heterozygous loss: embryonic lethality in high eukaryotes, cell cycle arrest Role in cancers: Oncogenic variants Regulate cell proliferation and metastasis (ER positive breast cancer, CRC, liver, lung, prostate, metastatic melanoma, PDAC and bladder)
H2A.Z.1	Transcription initiation and elongation	Mouse early embryonic development	Alterations’ consequences H2A.Z.1 KO: mouse early embryonic lethality, cell cycle arrest Role in cancers: Oncogenic variants Regulate tumorigenesis and metastasis (HCC and ICC)
H2A.Z.2	Gene regulation	Regulate cell proliferation and apoptosis	Alterations’ consequences H2A.Z.2 KO: Reduced cell proliferation but increase apoptosis, cell cycle arrest Role in cancers: Oncogenic variants Regulate cell proliferation and metastasis (aggressive melanoma, PDAC)
H2A.Z.2.2	Destabilizing nucleosome	Unknown	Alterations’ consequences H2A.Z.2.2 KO: Cell cycle arrest Role in cancers: Oncogenic variants Regulate cell proliferation and metastasis (PDAC)
MacroH2A1	Maintainer of heterochromatin architecture	1. Stabilizing XCI 2. Regulate memory processing and formation	Role in cancers: Tumor suppressor Regulate cell proliferation, migration, and metastasis (melanoma, breast, liver, lung, bladder, cervical, ovarian cancers, and CRC)
MacroH2A2	Maintainer of heterochromatin architecture and chromatin organization	1. Stabilizing XCI 2. Reprogramming barrier in fully differentiated cells	Role in cancers: Tumor suppressor Regulate cell proliferation, metastasis, and gene expressions
MacroH2A1.1	Maintainer of heterochromatin architecture	1. NAD+ metabolism 2. ADP-ribose signaling 3. DNA damage repair (NHEJ)	Role in cancers: Tumor suppressor Regulate cancer metastasis and cell proliferation
MacroH2A1.2	Maintainer of heterochromatin architecture	1. Inhibit PARP-1 enzymatic activity 2. Mediating homologous repair (HR)	Role in cancers: Tumor suppressor Regulate cell proliferation, inhibit cancers induced osteoclastogenesis
H2A.R	Spermatogenesis	Unknown	Unknown
H2A.Bbd	Transcriptional regulation	Controlling preimplantation embryonic development	Role in cancers: Oncogenic variants Cells with shorter S phase, increased sensitivity to DNA damage Regulate gene transcription regulation and cell proliferation (HL, BLCA, UCEC, cervical squamous cell carcinomas and endocervical carcinoma)
H2A.P	Unknown	Unknown	Role in cancers: Oncogenic variants (Unknown functions)
H2A.Q	Unknown	Unknown	Role in cancers: Oncogenic variants (Unknown functions)
H2A.L	Unknown	Unknown	Role in cancers: Oncogenic variants (Unknown functions)
H2A.L.2	Spermatogenesis, male mouse fertility	Unknown	Alterations’ consequences H2A.L.2 KO: complete mouse fertility Role in cancers: Oncogenic variants (Unknown functions)
H2A.L.3	Unknown	Unknown	Unknown
H2A.22 (H2A.J)	Stimulate inflammatory signaling cascade during DNA damage	Tissues or organs specific functions regardless of aging	Role in cancers: Oncogenic variants In luminal type B breast cancer, KIRC, aggressive melanoma, brain Role in cancers: Tumor suppressor In prostate, bladder cancers and all subtypes of breast cancer excluding luminal A Regulate gene expressions (ER positive breast cancer and prostate cancer) Mediate chemoradiotherapy resistance through signaling pathways (CRC, HCC, and glioblastoma)
H2A.1	Spermatogenesis, iPSCs generation and early embryogenesis	Unknown	Alterations’ consequences H2A.1 KD: perturbation in genome reprogramming Role in cancers: Oncogenic variants Regulate cell proliferation (HCC, CRC)
H2A.2	Mice neurons differentiation, embryogenesis, and aging	Unknown	Role in cancers Hyper-methylated in HCC (Unknown functions) Varying degree of PTMs

Abbreviations: mESCs: mouse embryonic stem cells; HCC: hepatocellular carcinoma; CRC: Colorectal cancer; hPSCs: human Pluripotent Stem Cells; ER: estrogen receptor; PDAC: pancreatic ductal adenocarcinoma; KO: knockout; ICC: intrahepatic cholangiocarcinoma; XCI: X chromosome inactivation; NHEJ: non-homologous end joining; HR: homologous repair; HL: Hodgkin’s lymphoma; BLCA: urothelial bladder carcinomas; UCEC: Uterine Corpus Endometrial Carcinomas; PARP-1: Poly(ADP-ribose) Polymerase 1; KIRC: Kidney Renal Cell Carcinoma; KD: knockdown; iPSCs: Induce Pluripotent Stem Cells; PTMs: post-transcriptional modifications.

**Table 3 ijms-25-03144-t003:** Histone variants’ tissues expression location, histone chaperones and remodelers and clinical/pathological importance as biomarker/prognostic indicator. This table summarized all the 19 H2A variant’s’ tissues expression location, their corresponding histone chaperones, or remodelers responsible for deposition or exchange in or out the chromatin, and their clinicopathological importance as a prognostic indicator or biomarker for different types of cancers at vary expression levels.

Variants’ Name	Location of Tissues	Chaperone and Remodeler	Clinicopathological Importance/Prognostic Value/Biomarker in Cancer Types
H2A.X	All tissues	FACT	High γ-H2A.X: 1. Biomarker for defect in DDR 2. Identified in cervical cancer, CRC, melanoma and OC 3. Access chemotherapy or radiotherapy effectiveness 4. Evaluate patient’s response to chemo-drug treatment and radiosensitivity 5. Predict high risk or early detection of cancers
H2A.Z	All tissues (pericentric heterochromatin in undifferentiated cells, TSS, centromere and enhancer regions)	ANP32E SRCAP INO80	High H2A.Z level: 1. Identified in breast cancer, CRC, liver, lung, prostate, bladder cancer, PDAC and metastatic melanoma 2. Poorer survival prediction 3. Therapeutic target for chemotherapy (cisplatin) 4. As biomarker for early PDAC and CRC detection
H2A.Z.1	All tissues (regulatory regions and heterochromatin)	ANP32E SRCAP INO80 P400	High H2A.Z.1 level: 1. Identified in breast tumor, PDAC and HCC 2. Indicator for lymph node metastasis and poorer survival prediction 3. Indicator for predicting which treatments patients respond the best
H2A.Z.2	All tissues (regulatory regions and heterochromatin)	P400 SRCAP	High H2A.Z.2 level: 1. Identified in PDAC 2. Not known for any prognostic and biomarker value
H2A.Z.2.2	All tissues (but predominantly in human brain)	TIP60 P400 SRCAP	High H2A.Z.2.2 level: 1. Identified in PDAC only 2. Not known for any prognostic and biomarker value
MacroH2A1	All tissues (Constitutive and facultative heterochromatin)	E3 ligase Cullin3^SPOP^	High macroH2A1 level: 1. Indicator of worst prognosis in different breast cancer types 2. Diagnostic indicator for HCC and lung cancer recurrence 3. Indicator for poor outcome prediction in HCC
MacroH2A2	All tissues (Constitutive and facultative heterochromatin)	Unknown	High macroH2A2 level: 1. Indicator of worst prognosis in different breast cancer types 2. Indicator of recurrence free survival in anal neoplasm Low macroH2A2 level: 1. Indicator of high-grade and faster recurrence of anal neoplasm
MacroH2A1.1	All tissues (Constitutive and facultative heterochromatin)	Unknown	High macroH2A1.1 level: 1. Biomarker for TNBC 2. Indicator of poorer survival prediction of TNBC 3. Biomarker for senescent cells in tumor Low macroH2A1.1 level: 1. Diagnostic indicator for lung cancer recurrence
MacroH2A1.2	All tissues (Constitutive and facultative heterochromatin)	ATRX	High macroH2A1.2 level: 1. Biomarker for HER2 positive breast cancer
H2A.R	Testis-specific	Unknown	Unknown
H2A.Bbd	All tissues (euchromatin and testis)	NAP1	High H2A.Bbd level: 1. Biomarker for HL, BLCA, UCEC, DLBCLs, etc. (required further research)
H2A.P	Originated on portion of X chromosome, testis-specific	Unknown	Unknown
H2A.Q	In testis of dogs and pigs only	Unknown	Unknown
H2A.L	Lost in human, conserved in mouse, testis-specific	Unknown	Unknown
H2A.L.2	Lost in human, testis-specific	Unknown	Unknown
H2A.L.3	Lost in human, testis-specific	Unknown	Unknown
H2A.22 (H2A.J)	In senescent cells’ chromatin, aging mice in tissue-specific manner and human skin	Unknown	High H2A.J level: 1. Poor prognostic marker in glioblastoma, KIRC, brain cancer and aggressive melanoma 2. Indicator for increasing survival rate in bladder, prostate and all subtypes of breast cancer 3. Biomarkers for senescent stem and aging skin cells 4. Indicator for cancer that exhibit chemotherapy resistance
H2A.1	X chromosome, autosomes, testis, oocytes and zygote	NPM2	High H2A.1 level: 1. Biomarker for HCC and colon cancer detection (required further research)
H2A.2	Unknown	Unknown	High H2A.2 level: 1. Biomarker for colon cancer detection (required further research)

Abbreviations: DDR: DNA damage repair; CRC: colorectal cancer; OC: ovarian cancer; HCC: hepatocellular carcinoma; FACT: Facilitate Chromatin Transcription; ANP32E: Acidic Nuclear Phosphoprotein 32 Family Member E; ANP32B: Acidic Nuclear Phosphoprotein 32B; INO80: human Inositol auxotrophy 80; SRCAP: Snf2 Related CREBBP Activator Protein; NAP1: Nucleosome Assembly Protein 1; TSS: transcription start site; PDAC: pancreatic ductal adenocarcinoma; TNBC: Triple Negative Breast Cancer; HER2: Human Epidermal Growth Factor Receptor 2; HL: Hodgkin’s lymphoma; BLCA: urothelial bladder carcinomas; UCEC: Uterine Corpus Endometrial Carcinomas; DLBCLs: Diffuse Large B-cell Lymphomas; KIRC: Kidney Renal Cell Carcinoma.

**Table 4 ijms-25-03144-t004:** Significance of the altered expression of histone variants in various cancer types. This table summarized the function of all 19 H2A variants according to the difference in their expression levels in cancer.

Variants’ Name	Altered Expression Levels Function in Cancer Types	
	Low Expression Levels	High Expression Levels
H2A.X	1. Upregulate key EMT-related transcription factors, Slug and ZEB1 → Induction of EMT and invasiveness 2. Hypoxic dependent H2A.X reduction will boost angiogenic and EGFR/HIF-1α/VEGF signaling pathway	1. Upregulation of PI3K/Akt/mTOR pathway → promote cell proliferation, escape apoptosis protein synthesis and extracellular matrix remodeling 2. Elevated Tyr39 phosphorylation of H2A.X → promote cell proliferation, metastasis and poorer cell differentiation
H2A.Z	/	1. Promote cell proliferation in ER positive breast cancer and prostate cancer; promote lymph node metastasis 2. ER positive breast cancer: upregulate ERα and c-Myc to enhance cell proliferation through increasing transcription of cell cycle genes
H2A.Z.1	1. Promote EMT through TGF-β signaling pathway	1. Suppressing cell apoptosis and negative cell cycle regulators 2. Upregulate cell cycle genes 3. Promote cell cycle progression in PDAC
H2A.Z.2	/	1. Upregulate E2F target genes’ transcription activities → promote melanoma progression and metastasis 2. Hyper-acetylation of H2A.Z.2 → increase binding with BRD2 to promote cell proliferation 3. Promote cell cycle progression in PDAC
H2A.Z.2.2	/	/
MacroH2A1	1. Promote cancer cell proliferation, migration and metastasis in melanoma	1. Effectively inhibit EMT induction to prevent metastasis
MacroH2A2	1. Promote cancer cell proliferation, migration and metastasis in melanoma	1. Repress self-renewal genes expressions 2. Impede metastasis in DCCs by inhibiting cell cycle and oncogenic signaling programs
MacroH2A1.1	1. Promote cancer cell proliferation, migration and metastasis in melanoma	1. Effectively inhibit EMT induction and prevent further mesenchymal morphology of cancer cells
MacroH2A1.2	1. Promote cancer cell proliferation, migration and metastasis in melanoma	1. Inhibit breast cancer induced osteoclastogenesis by repressing *LOX* gene through raising H3K27me3 level for gene silencing 2. Inhibit prostate cancer induced osteoclastogenesis by direct interactions with HP1α and H1.2, inactivate *LT**β* gene
H2A.R	/	/
H2A.Bbd	/	1. Cells with shorter S phase and promote cell proliferation in HL 2. Upregulate rDNA expression to promote cell proliferation
H2A.P	/	/
H2A.Q	/	/
H2A.L	/	/
H2A.L.2	/	/
H2A.L.3	/	/
H2A.22 (H2A.J)	/	1. Modulate estrogen and metastasis-regulated genes in ER-positive breast and prostate cancers 2. Causing chemoradiotherapy resistance in CRC through regulating MAPK7, HIV Nef pathway and inflammatory pathways 3. Causing chemotherapy resistance in HCC through accelerating PI3K/Akt and JAK/STAT; TNF-α/NF-κB, EMT and IL-6/STAT3 signaling pathway to control cell proliferation, migration and anti-apoptosis response 4. Causing radiotherapy and drug resistance in glioblastoma through activating TNF-α/NF-κB pathways, contacting with IL-6/STAT3 and HDAC3
H2A.1	/	1. Enhance gene transcription in HCC to promote cell proliferation 2. Activation of malignancy related genes 3. Hepatocytes reprogramming → further promote HCC development
H2A.2	/	1. Expressed in preneoplastic stages

Abbreviations: EMT: epithelial-to-mesenchymal transition; PI3K: Phosphoinositide 3-kinase; Akt: Protein kinase B; mTOR: Mammalian Target of Rapamycin; EGFR/HIF-1α/VEGF: Epidermal Growth Factor Receptor/Hypoxia-Inducible Factor 1 alpha/Vascular Endothelial Growth Factor; ER: estrogen receptor; ERα: estrogen receptor alpha; PDAC: pancreatic ductal adenocarcinoma; TGF-β: Transforming Growth Factor—beta; BRD2: Bromodomain-containing protein 2; DCC: Disseminated Cancer Cells; LOX: Lysyl Oxidase; HP1α: SUMOylated Heterochromatin Protein 1-Alpha; LTβ: Lymphotoxin Beta; CRC: Colorectal cancer; HCC: hepatocellular carcinoma; MAPK7: Mitogen-activated Protein Kinase 7; HIV: Human Immunodeficiency Virus; Nef: Negative Factor; JAK/STAT: Janus Kinase/Signal Transducers and Activators of Transcription; TNF-α/NF-κB: Tumor Necrosis Factor—Alpha/Nuclear Factor Kappa-light-chain-enhancer of activated B cells; IL-6/STAT3: Interleukin 6/Signal Transducers and Activators of Transcription 3; HDAC3: Histone Deacetylase 3.

## Data Availability

Data availability in a publicly accessible repository that does not issue DOIs: https://www.uniprot.org, https://www.ncbi.nlm.nih.gov (all accessed on 8 February 2024).

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
