# Peer review of "Roles of Histone H2A Variants in Cancer Development, Prognosis, and Treatment"

_ijms, 2024, doi:10.3390/ijms25063144_

Round 1

Reviewer 1 Report

Comments and Suggestions for Authors

This manuscript reviews the roles of histone H2A variants in cancer development, prognosis, and treatment. Authors summarize the functions of the 19 mammalian H2A variants and their roles in cancer biology. Authors provide the updated information regarding the roles of histone H2A variants in cancer development, prognosis, and treatment. The review is well organized and informative, but many grammatical and wording errors are detected and need to be improved. Many sentences are too long which may somehow confuse the readers. I recommend acceptance of this manuscript after language editing. 

Examples of wording errors:

- “substituite” should be “substitute”…line 8

- “histone 2A” should be “histone H2A”…line 13

- “testes specific” should be “testis specific”…line 16

- “specie specific traits” should be “species specific traits”…line 50

- “histone 2A” should be “histone H2A”…line 66

- “Additionaly” should be “Additionally”…line 74

- “histone 2A” should be “histone H2A”…line 107

- “differ” should be “different”…line 108

- “P13K” should be “PI3K”…line 133

- “reqquired” should be “required”…line 145

- “phosphorylaiton” should be “phosphorylation”…line 155

- “leadiang” should be “leading”…line 191

- “eukaryotes” should be “eukaryotic”…line 202

- “P13K” should be “PI3K”…line 216

- “cervix” should be “cervical cancer”…line 253

- “soley” should be “solely”…line 334

- “remained for elucidation” should be “remains to be elucidated”…line 367

- “in combine” should be “in combination”…line 442

- “inhibit” should be “inhibiting”…line 444

- “umour” should be “tumour”…line 546

- “compare” should be “compared”…line 660

- “have” should be “has”…line 691

- “Contrarily” should be “On the contrary”…line 790

- “while” should be “but on”…line 886

- “have” should be “has”…line 946

- “intergrity” should be “integrity”…line 965

- “expolited” should be “exploited”…line 969

- “fuunctional” should be “functional”…line 974

- “invovled” should be “involved”…line 979

Examples of sentences that should be re-written:

- It is known that somatic tissues are the main location for most variants’ expression yet, variants undergo further specialization and evolved to become lineage-specific according to the host organisms and exhibit high degree of variation across species, enabling them to carry out additional tasks, for example, some are predominantly expressed in male germline and participated in functions like sperm packaging 16. (lines 60-65)

- Importantly, downregulating of SPT16 lead to the accumulation of H2A.X at repair sites reduced markedly. (lines 144-145)

- Re-cently, acetylated H2A.Z has a novel contribution in androgen receptor (AR) dependent prostate cancer is its ability in generating new ectopic enhancers, resulting in aberrant genes transcriptional activation120. (lines 426-429)

- Recent reports also demonstrated that when inducing EMT in immortalized HMLE (Human Mammary Epithelial Cells), macroH2A1 expression dramatically decreased. (lines 554-555)

- In HER2 (Human Epidermal Growth Factor Receptor 2) positive breast cancer, macroH2A1.2 expression level is extremely high, with the presence of trinucleotide insertion (-EIS-) sequences, that are absent from macroH2A1.1, is responsible for the interaction and binding between HER2 receptors and its promoter153. (lines 600-604)

- As studies exploited that macroH2A1.1 is downregulated in carcinoma that already triumph over such an anti-tumor mechanisms and display uncontrollable cell proliferation. Thereby measuring the macroH2A1.1 level, patients with lower expression are more likely to have lung cancer recurrence compare to other isoforms that display no prognostic corelation towards tumour recurrence152. (lines 620-624)

- Detailed phylogenomic analyses have been done on this group of sH2A and recently, identified an undescribed variants, H2A.R, which is suggested to be a common ancestor for all the known sH2A up to date165,166. (lines 641-643)

- Previous work have shown that even though H2A.B tends to localize at active gene transcriptional regions, cells that express H2A.B are more prone to have a shorter S phase and an accelerated sensitivity towards DNA damage, which these are associated to oncogenesis165. (lines 691-694)

- Notably, H2A.2 gradually increases accompany with decreases of H2A.1 is observed in adult hepatocytes215. (lines 927-928)

Comments on the Quality of English Language

This manuscript reviews the roles of histone H2A variants in cancer development, prognosis, and treatment. Authors summarize the functions of the 19 mammalian H2A variants and their roles in cancer biology. Authors provide the updated information regarding the roles of histone H2A variants in cancer development, prognosis, and treatment. The review is well organized and informative, but many grammatical and wording errors are detected and need to be improved. Many sentences are too long which may somehow confuse the readers. I recommend acceptance of this manuscript after language editing. 

Examples of wording errors:

- “substituite” should be “substitute”…line 8

- “histone 2A” should be “histone H2A”…line 13

- “testes specific” should be “testis specific”…line 16

- “specie specific traits” should be “species specific traits”…line 50

- “histone 2A” should be “histone H2A”…line 66

- “Additionaly” should be “Additionally”…line 74

- “histone 2A” should be “histone H2A”…line 107

- “differ” should be “different”…line 108

- “P13K” should be “PI3K”…line 133

- “reqquired” should be “required”…line 145

- “phosphorylaiton” should be “phosphorylation”…line 155

- “leadiang” should be “leading”…line 191

- “eukaryotes” should be “eukaryotic”…line 202

- “P13K” should be “PI3K”…line 216

- “cervix” should be “cervical cancer”…line 253

- “soley” should be “solely”…line 334

- “remained for elucidation” should be “remains to be elucidated”…line 367

- “in combine” should be “in combination”…line 442

- “inhibit” should be “inhibiting”…line 444

- “umour” should be “tumour”…line 546

- “compare” should be “compared”…line 660

- “have” should be “has”…line 691

- “Contrarily” should be “On the contrary”…line 790

- “while” should be “but on”…line 886

- “have” should be “has”…line 946

- “intergrity” should be “integrity”…line 965

- “expolited” should be “exploited”…line 969

- “fuunctional” should be “functional”…line 974

- “invovled” should be “involved”…line 979

Examples of sentences that should be re-written:

- It is known that somatic tissues are the main location for most variants’ expression yet, variants undergo further specialization and evolved to become lineage-specific according to the host organisms and exhibit high degree of variation across species, enabling them to carry out additional tasks, for example, some are predominantly expressed in male germline and participated in functions like sperm packaging 16. (lines 60-65)

- Importantly, downregulating of SPT16 lead to the accumulation of H2A.X at repair sites reduced markedly. (lines 144-145)

- Re-cently, acetylated H2A.Z has a novel contribution in androgen receptor (AR) dependent prostate cancer is its ability in generating new ectopic enhancers, resulting in aberrant genes transcriptional activation120. (lines 426-429)

- Recent reports also demonstrated that when inducing EMT in immortalized HMLE (Human Mammary Epithelial Cells), macroH2A1 expression dramatically decreased. (lines 554-555)

- In HER2 (Human Epidermal Growth Factor Receptor 2) positive breast cancer, macroH2A1.2 expression level is extremely high, with the presence of trinucleotide insertion (-EIS-) sequences, that are absent from macroH2A1.1, is responsible for the interaction and binding between HER2 receptors and its promoter153. (lines 600-604)

- As studies exploited that macroH2A1.1 is downregulated in carcinoma that already triumph over such an anti-tumor mechanisms and display uncontrollable cell proliferation. Thereby measuring the macroH2A1.1 level, patients with lower expression are more likely to have lung cancer recurrence compare to other isoforms that display no prognostic corelation towards tumour recurrence152. (lines 620-624)

- Detailed phylogenomic analyses have been done on this group of sH2A and recently, identified an undescribed variants, H2A.R, which is suggested to be a common ancestor for all the known sH2A up to date165,166. (lines 641-643)

- Previous work have shown that even though H2A.B tends to localize at active gene transcriptional regions, cells that express H2A.B are more prone to have a shorter S phase and an accelerated sensitivity towards DNA damage, which these are associated to oncogenesis165. (lines 691-694)

- Notably, H2A.2 gradually increases accompany with decreases of H2A.1 is observed in adult hepatocytes215. (lines 927-928)

Author Response

Comments and Suggestions for Authors

This manuscript reviews the roles of histone H2A variants in cancer development, prognosis, and treatment. Authors summarize the functions of the 19 mammalian H2A variants and their roles in cancer biology. Authors provide the updated information regarding the roles of histone H2A variants in cancer development, prognosis, and treatment. The review is well organized and informative, but many grammatical and wording errors are detected and need to be improved. Many sentences are too long which may somehow confuse the readers. I recommend acceptance of this manuscript after language editing. 

Response:

Thank you very much for the insightful comments. We have revised the manuscript accordingly and have highlighted the changes in yellow in the revised version. We have also sent out the manuscript for professional language editing.

Examples of wording errors:

- “substituite” should be “substitute”…line 8

- “histone 2A” should be “histone H2A”…line 13

- “testes specific” should be “testis specific”…line 16

- “specie specific traits” should be “species specific traits”…line 50

- “histone 2A” should be “histone H2A”…line 66

- “Additionaly” should be “Additionally”…line 74

- “histone 2A” should be “histone H2A”…line 107

- “differ” should be “different”…line 108

- “P13K” should be “PI3K”…line 133

- “reqquired” should be “required”…line 145

- “phosphorylaiton” should be “phosphorylation”…line 155

- “leadiang” should be “leading”…line 191

- “eukaryotes” should be “eukaryotic”…line 202

- “P13K” should be “PI3K”…line 216

- “cervix” should be “cervical cancer”…line 253

- “soley” should be “solely”…line 334

- “remained for elucidation” should be “remains to be elucidated”…line 367

- “in combine” should be “in combination”…line 442

- “inhibit” should be “inhibiting”…line 444

- “umour” should be “tumour”…line 546

- “compare” should be “compared”…line 660

- “have” should be “has”…line 691

- “Contrarily” should be “On the contrary”…line 790

- “while” should be “but on”…line 886

- “have” should be “has”…line 946

- “intergrity” should be “integrity”…line 965

- “expolited” should be “exploited”…line 969

- “fuunctional” should be “functional”…line 974

- “invovled” should be “involved”…line 979

Response:

We have revised all the above wording errors. Those which have been changed are highlighted in yellow while some of the above changes have been removed after language editing.

Examples of sentences that should be re-written:

- It is known that somatic tissues are the main location for most variants’ expression yet, variants undergo further specialization and evolved to become lineage-specific according to the host organisms and exhibit high degree of variation across species, enabling them to carry out additional tasks, for example, some are predominantly expressed in male germline and participated in functions like sperm packaging 16. (lines 60-65)

- Importantly, downregulating of SPT16 lead to the accumulation of H2A.X at repair sites reduced markedly. (lines 144-145)

- Re-cently, acetylated H2A.Z has a novel contribution in androgen receptor (AR) dependent prostate cancer is its ability in generating new ectopic enhancers, resulting in aberrant genes transcriptional activation120. (lines 426-429)

- Recent reports also demonstrated that when inducing EMT in immortalized HMLE (Human Mammary Epithelial Cells), macroH2A1 expression dramatically decreased. (lines 554-555)

- In HER2 (Human Epidermal Growth Factor Receptor 2) positive breast cancer, macroH2A1.2 expression level is extremely high, with the presence of trinucleotide insertion (-EIS-) sequences, that are absent from macroH2A1.1, is responsible for the interaction and binding between HER2 receptors and its promoter153. (lines 600-604)

- As studies exploited that macroH2A1.1 is downregulated in carcinoma that already triumph over such an anti-tumor mechanisms and display uncontrollable cell proliferation. Thereby measuring the macroH2A1.1 level, patients with lower expression are more likely to have lung cancer recurrence compare to other isoforms that display no prognostic corelation towards tumour recurrence152. (lines 620-624)

- Detailed phylogenomic analyses have been done on this group of sH2A and recently, identified an undescribed variants, H2A.R, which is suggested to be a common ancestor for all the known sH2A up to date165,166. (lines 641-643)

- Previous work have shown that even though H2A.B tends to localize at active gene transcriptional regions, cells that express H2A.B are more prone to have a shorter S phase and an accelerated sensitivity towards DNA damage, which these are associated to oncogenesis165. (lines 691-694)

- Notably, H2A.2 gradually increases accompany with decreases of H2A.1 is observed in adult hepatocytes215. (lines 927-928)

Response:

We have re-written all the above sentences/paragraph and highlighted in yellow. Thank you very much for the comments and suggestions.

Reviewer 2 Report

Comments and Suggestions for Authors

Lai et al. focused on the respective functions of H2A variants in cancer biology and clinical setting. Nineteen mammalian H2A variants were divided into five H2A.X-Z, four macroH2A, 7 short H2A and three others, all of which functions and clinical importance were explained in this review article. Now, alterations of histone H2A have been highlighted in cancer development, and several review articles have been reported. Therefore, this is an interesting topic and informative review article. However, explanation of each H2A variants are too long and the Figures are very simply, leading us that this review article seems to be difficult to read and understand. 

Comments; 

1)    The authors should add the new tables summarizing clinicopathological importance in H2A altered cancer. For instance, a) clinical importance (clinicopathological importance, biomarker etc.) should be summarized within additional one Tables. b) alterations and functions of histone H2A should be summarized within additional one Tables. Although this review article did not cite the Buschbeck’s review article for variants of core histone, this article may brilliantly summarize it using excellent figures and tables (but lacks the explanation of the importance as a biomarker). c) because there are a few reviews summarizing H2A for cancer detection, the H2A alterations as a useful biomarker should be summarized in new table. 

2)    I think that Figures 2, 3, 5, 6 (very simple function) do not give us useful information. What do the authors want to strengthen using these Figures? In the way of Figure 4, the figures for alteration of H2A are very important to understand its function in cancer cells. If the authors would like to use them, these Figures should be reconstructed.

Author Response

Reviewer #2

Comments and Suggestions for Authors

Lai et al. focused on the respective functions of H2A variants in cancer biology and clinical setting. Nineteen mammalian H2A variants were divided into five H2A.X-Z, four macroH2A, 7 short H2A and three others, all of which functions and clinical importance were explained in this review article. Now, alterations of histone H2A have been highlighted in cancer development, and several review articles have been reported. Therefore, this is an interesting topic and informative review article. However, explanation of each H2A variants are too long and the Figures are very simply, leading us that this review article seems to be difficult to read and understand. 

Response:

Thank you very much for the positive and useful comments and suggestions. We have revised/rephrased some of the paragraphs accordingly and the manuscript has been further revised by professional language editing. 

Comments; 

1)    The authors should add the new tables summarizing clinicopathological importance in H2A altered cancer. For instance, a) clinical importance (clinicopathological importance, biomarker etc.) should be summarized within additional one Tables. b) alterations and functions of histone H2A should be summarized within additional one Tables. Although this review article did not cite the Buschbeck’s review article for variants of core histone, this article may brilliantly summarize it using excellent figures and tables (but lacks the explanation of the importance as a biomarker). c) because there are a few reviews summarizing H2A for cancer detection, the H2A alterations as a useful biomarker should be summarized in new table. 

Response:

Thank you very much for the suggestions. We have added Table 2, a new table summarizing the functions, alterations and role in cancers of all variants of histone H2A for suggestion “b)” mentioned in your initial comment. We have added another table: “Table 3” summarizing the location of expression of histone H2A variants, their chaperones and remodellers and clinical/pathological importance as biomarker/prognostic indicator and Table 4 is added to summarize the significance of the altered expression of histone variants in various cancer for suggestions “a) and c)”.  

2)    I think that Figures 2, 3, 5, 6 (very simple function) do not give us useful information. What do the authors want to strengthen using these Figures? In the way of Figure 4, the figures for alteration of H2A are very important to understand its function in cancer cells. If the authors would like to use them, these Figures should be reconstructed.

Response:

Thank you for the suggestion. We have revised the original Figure 2 showing the role of overexpression and down-regulation of H2A.X in different types of cancers. We have removed the original Figure 3 and kept the original figure 4 (changed to revised figure 3). The original Figure 5 has been revised and strengthened accordingly and changed to revised Figure 4. We think that the original figure 6 (revised to be figure 5) is required to better illustrate the functional interplay between histone, H2A.L.2, Transition proteins and Protamines during spermatogenesis in mice which would be hard to fully understand by just reading the main text. We have additionally add a new figure 6, summarizing all the signalling pathways involved in how overexpression of H2A.J lead to drug resistance in cancer therapy.

Round 2

Reviewer 1 Report

Comments and Suggestions for Authors

The authors have addressed all the points of concern.

Author Response

Thank you.

Reviewer 2 Report

Comments and Suggestions for Authors

The authors have addressed the issues I mentioned in the review. 

To improve the manuscript, I would like to add only one minor comment.

Explanation of H2A.X for EMT (Slug and ZEB) and gamma H2A.X for EFGR pathway (page 10 lines 222-231) are summarized in Table 4 (although the authors did not add the in the sentence). However, Table 3 is firstly appeared in page 12, line 13, and so the readers have confusion of where to look. To avoid the confusion, the authors should modify the sentence of “----cancer progression and metastasis (Table 2)” to “----cancer progression and metastasis (Tables 2-4)” in page 9, line 200.

Author Response

Response:

Thank you very much for the comment. We have revised the manuscript according to your suggestion. Please see the revised version of the changes highlighted in yellow (line 237, page 10). Thank you.